# Roles of the membrane-binding motif and the C-terminal domain of RNase E in localization and diffusion in *E. coli*

Laura Troyer[1†], Yu-Huan Wang[1†], Shobhna Shobhna[2,3], Seunghyeon Kim[1], Brooke Ramsey[4], Jeechul Woo[5], Emad Tajkhorshid[2,3,4]*, Sangjin Kim[1,4]*

[1]Department of Physics, University of Illinois Urbana-Champaign, Urbana, United States; [2]Theoretical and Computational Biophysics Group, NIH Center for Macromolecular Modeling and Visualization, Beckman Institute for Advanced Science and Technology, University of Illinois Urbana-Champaign, Urbana, United States; [3]Department of Biochemistry, University of Illinois Urbana-Champaign, Urbana, United States; [4]Center for Biophysics and Quantitative Biology, University of Illinois Urbana–Champaign, Urbana, United States; [5]Moduli Technologies, LLC, Springfield, United States

**\*For correspondence:**
emad@illinois.edu (ET);
sangjin@illinois.edu (SK)

[†]These authors contributed equally to this work

## eLife Assessment

This **valuable** study uses single-molecule imaging to characterize factors controlling the localization, mobility, and function of RNase E in *E. coli*, a key bacterial ribonuclease central to mRNA catabolism. The supporting evidence for the differential roles of RNAse E's membrane targeting sequence (MTS) and the C-terminal domain (CTD) to RNAse E's diffusion and membrane association is **convincing**. It provides insight into how RNAse E shapes the spatiotemporal organization of RNA processing in bacterial cells. This interdisciplinary work will be of interest to cell biologists, microbiologists, biochemists, and biophysicists.

**Abstract** In *Escherichia coli*, RNase E, a central enzyme in RNA processing and mRNA degradation, contains a catalytic N-terminal domain, a membrane-targeting sequence (MTS), and a C-terminal domain (CTD). We investigated how MTS and CTD influence RNase E localization, diffusion, and function. Super-resolution microscopy revealed that ~93% of RNase E localizes to the inner membrane and exhibits slow diffusion similar to polysomes. Comparing the native amphipathic MTS with a transmembrane motif showed that the MTS confers slower diffusion and stronger membrane binding. The CTD further slows diffusion by increasing mass but unexpectedly weakens membrane association. RNase E mutants with partial cytoplasmic localization displayed enhanced co-transcriptional degradation of *lacZ* mRNA. These findings indicate that variations in the MTS and the presence of the CTD shape the spatiotemporal organization of RNA processing in bacterial cells, providing mechanistic insight into how RNase E domain architecture influences its cellular function.

## Introduction

RNase E (RNE) is the main endoribonuclease in *Escherichia coli*, known for its role in RNA processing and mRNA degradation (*Mudd et al., 1990*; *Mackie, 2013*; *Hui et al., 2014*). It is an essential protein (*Apirion and Lassar, 1978*; *Babitzke and Kushner, 1991*), and homologous proteins are found across many bacterial species (*Carpousis, 2007*; *Aït-Bara and Carpousis, 2015*; *Mardle et al.,*

*2019*). The essentiality stems from the N-terminal domain (NTD), or the catalytic domain (*Callaghan et al., 2005*). The NTD is followed by a membrane-targeting sequence (MTS) and the C-terminal domain (CTD), or macromolecular interaction domain (*Mackie, 2013*), where RhlB (a DEAD-box RNA helicase), PNPase (a 3′ → 5′ exonuclease), and enolase (a glycolytic enzyme) bind to form the RNA degradosome complex (*Carpousis, 2007*). The MTS forms an amphipathic α helix, responsible for the localization of RNE on the inner membrane (*Khemici et al., 2008*; *Strahl et al., 2015*). Interestingly, the membrane localization of RNE and the presence of the CTD are not essential in *E. coli* nor are they conserved across bacterial species, in contrast to the broad conservation of the NTD across bacteria as well as chloroplasts (*Aït-Bara and Carpousis, 2015*; *Kaberdin et al., 1998*; *Lee and Cohen, 2003*). This raises a question about the roles of membrane localization and the CTD in the in vivo function of RNE.

*E. coli* strains with cytoplasmic RNE (due to the removal of the MTS) are viable, although they grow more slowly than the wild-type (WT) cells (*Khemici et al., 2008*; *Hadjeras et al., 2019*). In vitro studies have shown that membrane binding of RNE does not necessarily increase its enzymatic activity (*Khemici et al., 2008*; *Hadjeras et al., 2019*). However, membrane localization is likely important for gene regulation in vivo because RNE becomes sequestered from the cytoplasmic pool of mRNAs, giving mRNAs time for translation. This idea is supported by our recent observation that the membrane-bound RNE limits the degradation of nascent mRNAs while cytoplasmic RNE (ΔMTS) can degrade nascent mRNAs during transcription (*Kim et al., 2024*). We found that transcripts encoding membrane proteins can be an exception to this rule, in that they can experience co-transcriptional degradation assisted by the transertion effect (*Kim et al., 2024*). These findings agree with results from a genome-wide study, indicating that the membrane localization of RNE allows for differential regulation of mRNA stability for genes encoding cytoplasmic proteins versus inner membrane proteins in *E. coli* (*Moffitt et al., 2016*).

Previous studies have reported evidence that *E. coli* RNE can localize in the cytoplasm—for example, when cells were grown anaerobically (*Murashko and Lin-Chao, 2017*) or when membrane fluidity was reduced by changes in lipid composition (*Gohrbandt et al., 2022*). These findings imply that RNE can dissociate from the membrane; however, the origin of its weak membrane binding remains unknown.

Across bacteria, several species within α-proteobacteria have cytoplasmic RNE (*Al-Husini et al., 2018*) while other species possess membrane-bound RNE. Among these, *B. subtilis* RNase Y (a functional homolog of RNE) associates with the membrane via a transmembrane (TM) motif (*Lehnik-Habrink et al., 2011*), instead of an amphipathic motif used by *E. coli* and other γ-proteobacteria (*Aït-Bara and Carpousis, 2015*). Given the diversity of membrane-binding motifs that have arisen through evolution, it should be possible to engineer *E. coli* RNE with a TM motif. Such a mutant would provide a useful model for investigating the impact of membrane-binding motifs on the localization, diffusion, and activity of RNE.

Lastly, the CTD of *E. coli* RNE is an intrinsically disordered region (*Callaghan et al., 2004*) that, while nonessential for cell viability, enhances the enzymatic activity of the NTD (*Lopez et al., 1999*; *Leroy et al., 2002*; *Islam et al., 2021*; *Kim et al., 2024*). As the primary binding site for degradosome components (*Carpousis, 2007*), the CTD is thought to facilitate mRNA degradation by recruiting these proteins near the catalytic NTD. However, our recent study showed that these associated proteins have minimal impact on the degradation rate of *lacZ* mRNA, whereas deletion of the CTD markedly stabilizes the transcript (*Kim et al., 2024*), suggesting a possible intramolecular allosteric effect within RNE. In the present study, we further show that the CTD modulates the membrane-binding affinity of RNE, possibly by affecting its conformation. These results reveal a previously underappreciated role of the CTD in regulating RNE function beyond degradosome assembly, with implications for how the spatial organization and structural dynamics of RNE fine-tune RNA degradation in bacteria.

In this study, we quantified the membrane-binding percentage (MB%) of RNE in *E. coli* using single-molecule microscopy and showed that membrane association governs its diffusion and mRNA degradation activity. Perturbing the native MTS, substituting it with LacY TM segments, and deleting the CTD collectively revealed how the MTS and CTD set RNE's spatial organization and provided routes for tuning activity through subcellular control.

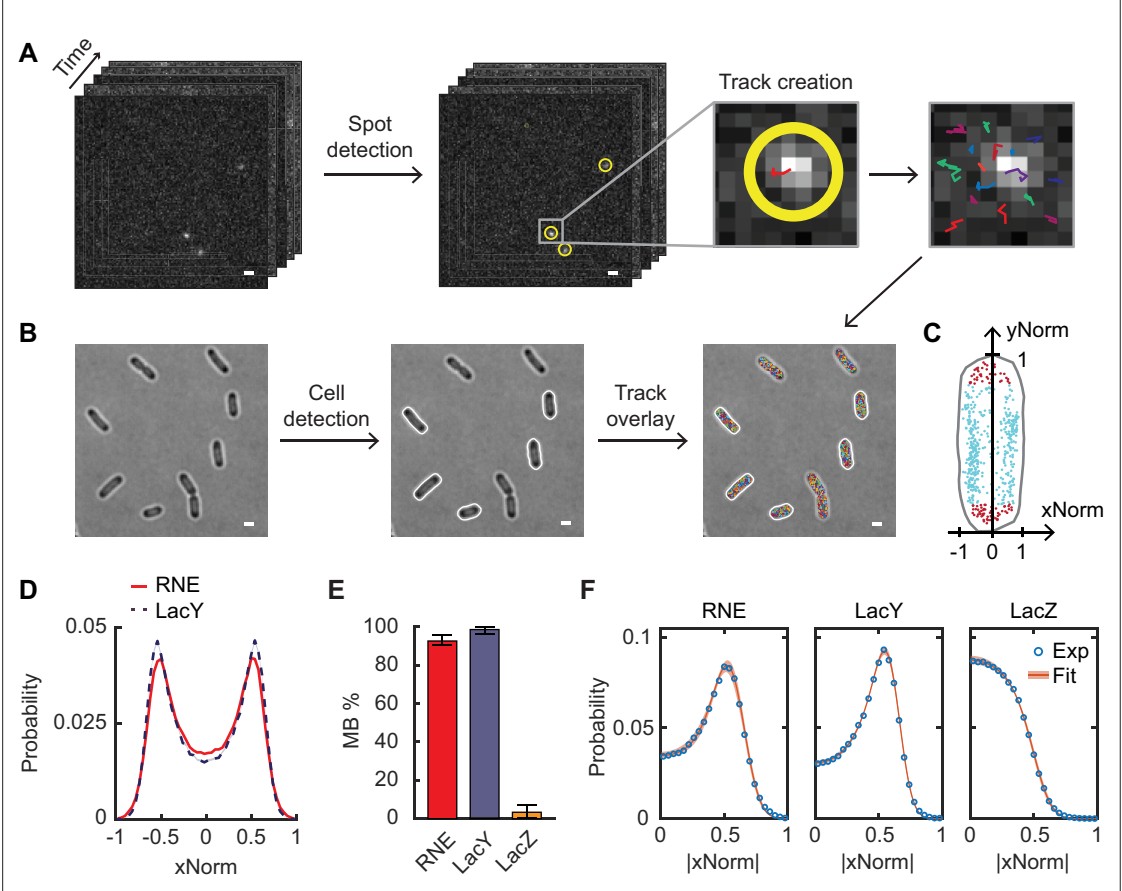

**Figure 1.** Analysis of single-molecule images for the subcellular localization and dynamics of proteins. (**A**) Single-molecule image analysis. Spots were detected in each frame (highlighted with yellow circles), and tracks were created across frames (different colors were chosen for different tracks). (**B**) Cell detection. Cell outlines were determined from bright-field images. Only non-dividing cells were analyzed (indicated by white outlines). (**C**) Normalized position of spots of RNE along the short (*x*) and the long (*y*) axes of an example cell. Red spots are inside the cell endcaps, and cyan spots are in the cylindrical region of the cell. (**D**) xNorm histogram of RNE and LacY. Only spots in the cylindrical region of cells (like cyan spots in **C**) were included, totaling $n$ = 143,000 spots. The standard error of the mean (SEM) calculated from bootstrapping is displayed as a shaded area but is smaller than the line width (see *Figure 1—figure supplement 1* for details). (**E**) The membrane-binding percentage (MB%) of RNE, LacY, and LacZ. Error bars are from the 95% confidence interval. (**F**) Histogram of absolute xNorm and model fitting of RNE, LacY, and LacZ to determine MB%. Orange highlights indicate the range of xNorm expected based on the standard deviations in the parameter values estimated by MCMC. The white scale bars in panels **A and B** are 1 μm. See *Supplementary file 6* for data statistics.

The online version of this article includes the following figure supplement(s) for figure 1:

**Figure supplement 1.** The xNorm histogram of RNE.

**Figure supplement 2.** Comparison of trajectory counts in live and fixed cells to test for undercounting of cytoplasmic molecules.

**Figure supplement 3.** xNorm fitting model.

## Results

### MB% of RNE

First, we investigated the subcellular localization of RNE in live cells. Previous fluorescence micros-copy studies have shown that RNE is localized to the inner membrane in *E. coli* (*Khemici et al., 2008*; *Strahl et al., 2015*; *Moffitt et al., 2016*), but the percentage of membrane-bound molecules has not been quantitatively examined in live cells. To address this gap, we fused RNE with a photo-convertible fluorescent protein, mEos3.2 (*Zhang et al., 2012*) and imaged individual RNE molecules over time in two dimensions (*Figure 1A*). The positions of fluorescent molecules were identified in each frame and linked into trajectories using the open-source software u-track (*Jaqaman et al., 2008*; *Figure 1A*). In this section, we analyze localization, and diffusion dynamics are addressed in subsequent sections.

Subcellular locations were calculated relative to the cell boundaries identified from bright-field images using another open-source image analysis package Oufti (*Paintdakhi et al., 2016*; *Figure 1B*). To combine data from many cells, molecular positions along the short and long axes of a cell were normalized to the cell width and cell length, yielding xNorm and yNorm, respectively (*Figure 1C*). Based on yNorm, molecules within the cylindrical part of the cell were selected, and their xNorm values were used to generate an xNorm histogram. Hereinafter, we focus on the xNorm histogram to compare the membrane enrichment across protein constructs.

The xNorm histogram of RNE shows two peaks corresponding to the inner membrane on each side of the cell (*Figure 1*, *Figure 1—figure supplement 1*), very similar to the xNorm histogram of LacY, obtained by imaging LacY-mEos3.2 using the same method. LacY is a membrane channel for lactose and is composed of 12 TM segments (*Abramson et al., 2003*). It is expected to be inserted into the inner membrane during translation (*Ahrem et al., 1989*; *Nagamori et al., 2003*; *Stochaj and Ehring, 1987*), such that all imaged LacY is expected to be localized in the inner membrane (*Volkov et al., 2022*).

We further quantified the percentage of molecules bound to the membrane (MB%) from the xNorm histogram. For this analysis, we first confirmed that proteins localized on the membrane and in the cytoplasm are detected with equal probability, despite differences in their mobilities (*Figure 1—figure supplement 2*). Next, we developed a mathematical model based on a 2D projection of molecules randomly distributed either on the surface of or within a cylinder. The model includes imaging effects that affect the shape of xNorm histograms: localization error, the limited focal depth of the quasi-TIRF illumination we used, and the location of the inner membrane relative to the cell boundary (*Figure 1—figure supplement 3*). Model fitting was performed using a Markov-Chain Monte Carlo algorithm (MCMC). For validation, we applied the model to xNorm histograms of LacY and LacZ, which serve as benchmarks for complete membrane binding and complete cytoplasmic localization, respectively. The MB% of LacY was 99% with a 95% confidence interval of [96%, 100%], and LacZ showed MB% of 3.4% [0.2%, 7.3%] (*Figure 1E, F*). Both MB% values agree with the expectations for membrane and cytoplasmic proteins. For RNE, we found an MB% of 93% [91%, 96%] (*Figure 1E*).

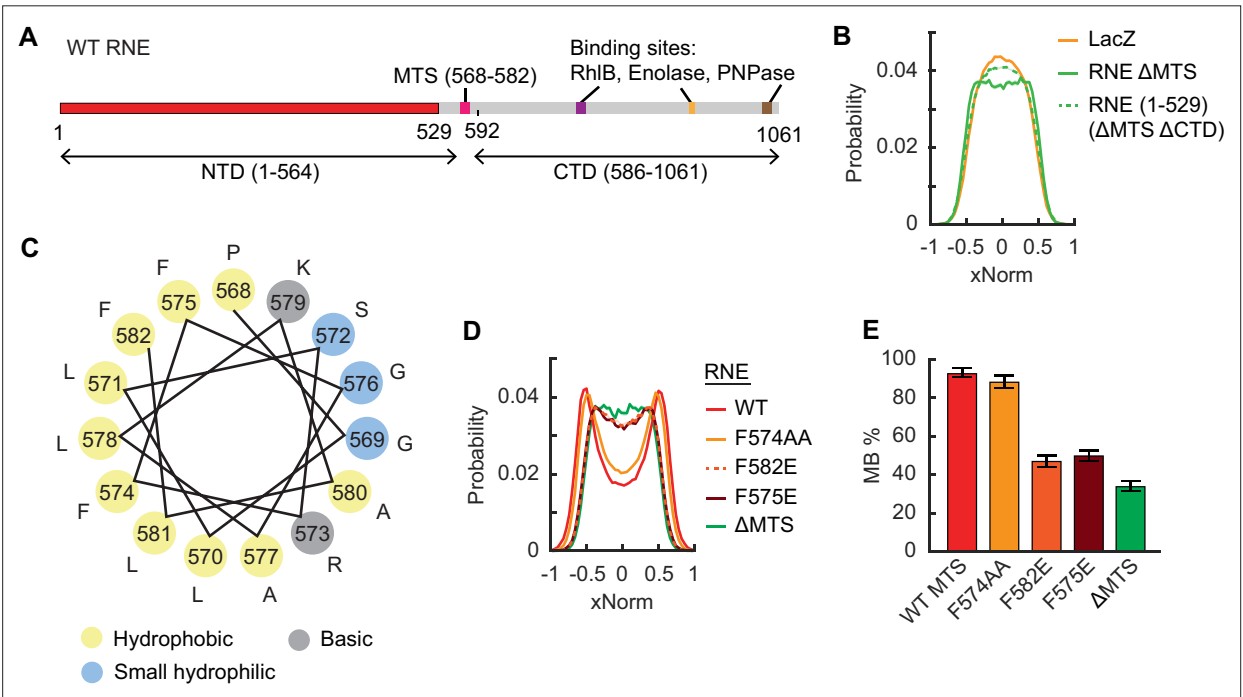

**Figure 2.** Mutations in the membrane-targeting sequence (MTS) affecting the localization of RNE. (**A**) Linear representation of the RNE monomer. The N-terminal domain (NTD) and the C-terminal domain (CTD) are defined as regions flanking the MTS. Numbers indicate the amino acid residues. (**B**) xNorm histograms of cytoplasmic RNE mutants. (**C**) Helical wheel diagram of the MTS region of RNE (residue 568–582). (**D**) xNorm histogram of RNE MTS point mutants. (**E**) Membrane-binding percentage (MB%) of RNE MTS point mutants. Error bars indicate the 95% confidence interval. In panels B and D, the SEM from bootstrapping is shown but is smaller than the line width. See *Supplementary file 6* for data statistics.

The xNorm histogram of the fastest 7% of the RNE population (based on the diffusion coefficient, as discussed below) exhibited a cytoplasmic localization pattern (without the two membrane-associated peaks), supporting the existence of a cytoplasmic RNE subpopulation (*Appendix 1—figure 2*).

We confirmed that the MTS (residues 568–582) is essential for the membrane binding of RNE, as deletion of the MTS sequence made the xNorm like that of LacZ (*Figure 2A, B*, *Appendix 1—figure 2*). We note that the xNorm profile of RNE ΔMTS was slightly different from that of LacZ near the center line of the cell (*x* = 0), suggesting fewer RNE ΔMTS molecules were at the midline of the cell. Also, xNorm fitting yielded an MB% of 33%. These results may reflect nucleoid exclusion of RNE ΔMTS due to its large size (*Thappeta et al., 2024*). Consistently, a smaller cytoplasmic RNE variant generated by CTD truncation (RNE (1–529) or RNE ΔMTS ΔCTD) exhibited a xNorm profile closer to that of LacZ (*Figure 2B*, *Appendix 1—figure 2*).

Is there a critical residue(s) within the MTS required for membrane binding? The MTS forms an amphipathic α-helix, in which hydrophobic residues are expected to align on one side of the helix (*Figure 2C*; *Khemici et al., 2008*). Previous studies suggested that replacing one of the hydrophobic residues with a hydrophilic amino acid can disrupt membrane binding of RNE (*Khemici et al., 2008*; *Strahl et al., 2015*). We revisited the point mutations discussed in *Khemici et al., 2008* and analyzed MB% by mEos3.2 imaging. We found that the F574A F575A double mutation (noted as F574AA) did not significantly affect MB%, consistent with the fact that the substituted amino acids remain hydrophobic (*Figure 2D, E*, *Appendix 1—figure 2*). In contrast, F582E and F575E mutations reduced

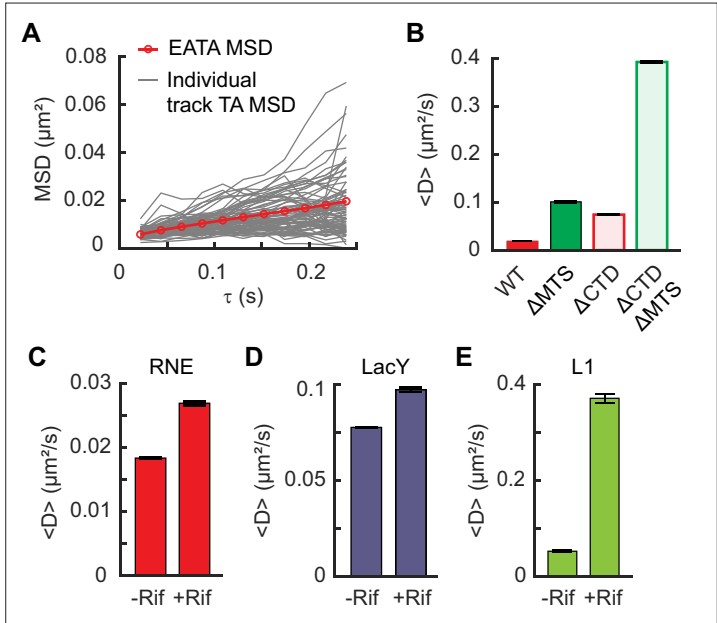

**Figure 3.** RNE diffusion, influenced by membrane binding and interactions with mRNAs. (**A**) Mean-squared displacement (MSD) versus time delay (*τ*) for RNE. Ensemble-averaged time-averaged (EATA) MSD was calculated by averaging the time-averaged MSD of individual tracks. (**B**) Mean diffusion coefficients of various RNE mutants, lacking the membrane-targeting sequence (MTS) and/or the C-terminal domain (CTD). Change in the mean diffusion coefficient of RNE (**C**), LacY (**D**), and ribosome L1 protein (**E**) when cellular RNAs were depleted by rifampicin treatment. Error bars in panels B–E represent the SEM. See ***Supplementary file 6*** for data statistics.

The online version of this article includes the following figure supplement(s) for figure 3:

**Figure supplement 1.** Distribution of RNE's diffusion coefficients and single- or double-Gaussian fit (both $R^2$ = 0.98).

**Figure supplement 2.** Ensemble-averaged, time-averaged (EATA) mean-squared displacement (MSD) of WT RNE, RNE in rif-treated cells, and ribosomal protein L1.

**Figure supplement 3.** Distribution of ribosomal protein L1's diffusion coefficients and a two-population Gaussian fit ($R^2$ = 0.99).

**Figure supplement 4.** Effect of chloramphenicol treatment and induction of lacZ mRNA from plasmids on RNE diffusion.

MB% to 47% and 50%, respectively (*Figure 2E*), and their xNorm histograms resembled that of RNE ΔMTS, suggesting predominantly cytoplasmic localization (*Figure 2D*, *Appendix 1—figure 2*). These findings indicate that the phenylalanine residues at positions 575 and 582 are critical for membrane association of RNE.

## Effect of membrane binding on the diffusion of RNE

The membrane localization of RNE likely limits its diffusion and interaction with mRNA targets. Additionally, interaction with ribosome-bound mRNAs can further slow the diffusion of RNE. Here, we examined how these factors contribute to the diffusion dynamics of RNE.

To measure the diffusion of RNE, we analyzed the trajectories of individual RNE-mEos3.2 imaged at a 21.7-ms acquisition interval (*Figure 1A*). We calculated the diffusion coefficient $D$ by fitting the mean-squared displacement (MSD) of each trajectory to the equation $MSD = 4D\tau + b$, where $\tau$ is lag time and $b$ accounts for both dynamic and static localization errors (*Savin and Doyle, 2005*; *Figure 3A*). We obtained $D_{RNE} = 0.0184 \pm 0.0002$ μm²/s (mean ± SEM, *Figure 3—figure supplement 1*). This value was well above the lower detection limit of our microscope, determined using stationary, surface-immobilized mEos3.2, $D = 0.0020 \pm 0.0001$ μm²/s (*Figure 3—figure supplement 2*). Notably, $D_{RNE}$ was comparable to that of ribosome-bound mRNAs, estimated to be $D \sim 0.015$ μm²/s based on the diffusion of ribosomal protein L1 (*Figure 3—figure supplement 3*).

To assess how membrane association affects diffusion, we compared $D$ of WT RNE and the ΔMTS mutant. These two proteins have similar molecular masses, as the MTS comprises only 15 of the 1061 residues in a monomer of RNE (*Figure 2A*). Therefore, any difference in $D$ can be attributed to their subcellular localizations (membrane vs. cytoplasm) rather than mass. We found that $D_{\Delta MTS}$ is ~5.5 times that of $D_{RNE}$ (*Figure 3B*).

We examined another pair of RNE mutants that differ in localization (membrane vs. cytoplasm) but are similar in size: membrane-bound RNE (1–592) and cytoplasmic RNE (1–529), both lacking the CTD. These truncated variants diffused faster than their full-length counterparts due to reduced mass, but their $D$ values still differed by a factor of ~5.3 due to localization (*Figure 3B*). Together, these results suggest that the membrane binding reduces RNE mobility by a factor of 5.

## Diffusion of RNE in the absence of mRNA substrates

When some of the RNE molecules interact with mRNA, their diffusion can slow due to the added mass of mRNA and ribosomes, possibly yielding slower RNE subpopulations. Such mobility-based subpopulations have been observed for RNA polymerases and ribosomes in *E. coli* and used to estimate the fraction of molecules interacting with RNA (*Bakshi et al., 2012*; *Sanamrad et al., 2014*; *Stracy et al., 2015*).

To test the effect of mRNA substrates on RNE diffusion, we treated cells with rifampicin (rif), which blocks transcription initiation, thus depleting cellular mRNAs (*Mosteller and Yanofsky, 1970*). In rif-treated cells, $D_{RNE}$ increased to $0.0270 \pm 0.0003$ μm²/s, which is $1.47 \pm 0.010$ times that in untreated cells (*Figure 3C*). We note that $D_{LacY}$ also increased to $1.25 \pm 0.01$ times relative to untreated cells (*Figure 3D*). Although this increase is relatively small, the increase in $D_{LacY}$ was unexpected because LacY is not an RNA-binding protein. The increase in $D_{LacY}$ likely results from the depletion of mRNAs near the membrane (e.g., the mRNAs undergoing transertion), which could otherwise hinder the diffusion of membrane proteins (*Binenbaum et al., 1999*; *Matsumoto et al., 2015*). The similar fold change in $D_{RNE}$ and $D_{LacY}$ upon rif treatment suggests that the change in RNE diffusion may largely be attributed to physical changes in the intracellular environment (such as reduced viscosity or macromolecular crowding; *Bellotto et al., 2022*; *Linnik et al., 2024*), rather than a loss of RNA–RNE interactions.

Because the rif-induced change in $D_{RNE}$ is largely physical, we next examined why eliminating RNA–RNE interactions does not further increase RNE mobility, using the ribosome as a benchmark. The diffusion of ribosomal protein L1 became seven times as fast as that in rif-treated cells (*Figure 3E*), consistent with previous reports (*Sanamrad et al., 2014*; *Bakshi et al., 2012*). This big change can be explained by the fact that ribosomes form polysomes, where the effective mass of L1 protein in untreated cells would be 2 or more times that in rif-treated cells (where it remains as a free subunit). In the case of RNE, it forms the RNA degradosome complex, whose mass can be from 450 kDa (*Carpousis, 2007*) to 2.3 MDa depending on the occupancy of the RNA degradosome component proteins (RhlB, PNPase, and enolase) (*Chandran and Luisi, 2006*; *Chandran et al., 2007*; *Mackie,*

*2013*; *Nurmohamed et al., 2010*; *Nurmohamed et al., 2009*) (see Appendix 2). Even at its largest size, the RNE complex is smaller than a 70S ribosome (~2.5 MDa; *Stark et al., 1995*). This means that if RNE interacts with an mRNA associated with n ribosomes, the total mass of the RNE complex would increase by a factor of *n* or more. Thus, a substantial increase in $D_{RNE}$ would be expected upon mRNA depletion, assuming that a significant fraction of RNE is engaged with mRNA–ribosome assemblies.

To explain the marginal increase in $D_{RNE}$ upon rif treatment, we considered two possibilities: (1) only a small percentage of RNE molecules interacts with mRNAs at a given time and/or (2) RNE interacts with mRNAs only briefly, unlike ribosomes which spend an order of 10–100 s in a polysome state during translation elongation (*Dai et al., 2016*). To distinguish between these possibilities, we attempted to increase the cellular pool of polysomes, either by treating cells with a translation elongation inhibitor chloramphenicol (*Stracy et al., 2015*) or by overexpressing *lacZ* mRNA from a high-copy plasmid (*Figure 3—figure supplement 4*). In both cases, a larger fraction of RNE would engage with mRNAs, potentially increasing the fraction of RNE in the slow-diffusing state. However, $D_{RNE}$ remained unchanged compared to untreated cells (*Figure 3—figure supplement 4*). This result rules out the possibility that only a small percentage of RNE interacts with mRNAs and instead weighs in favor of the scenario that RNE–mRNA interactions are brief. Specifically, if RNE interacts with mRNAs for ~20 ms or less, the slow-diffusing state would last shorter than the frame interval and remain undetected in our experiment.

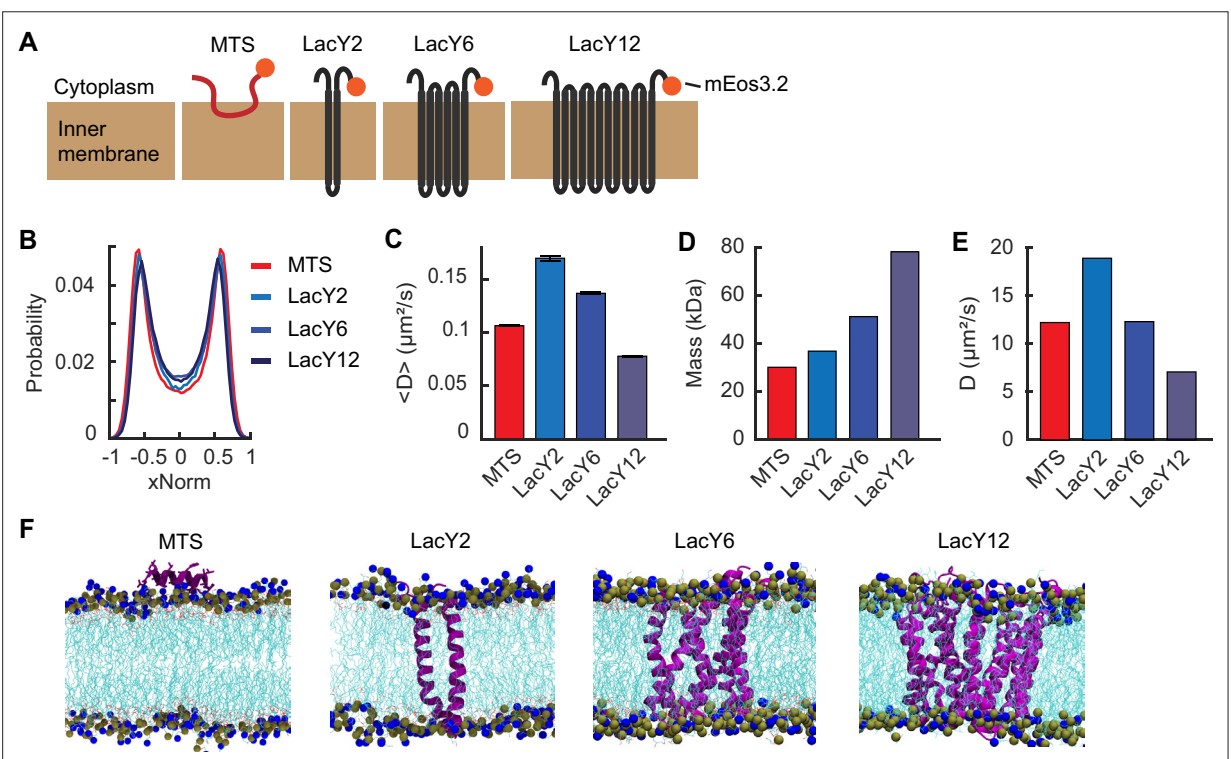

**Figure 4.** Localization and diffusion of membrane-binding motifs. (**A**) Cartoon schematic of the membrane-binding motifs used in this study (not to scale). The orange circles indicate mEos3.2 used for imaging. (**B**) xNorm histograms of membrane-binding motifs. The SEM from bootstrapping is displayed but smaller than the line width. Data are from at least 107,000 spots. (**C**) Mean diffusion coefficients of membrane-binding motifs. Error bars are the SEM from at least 3000 tracks. (**D**) Estimated mass of membrane-binding motifs based on the amino acid sequence including linkers and mEos3.2. (**E**) Diffusion coefficients of the membrane-binding motifs obtained from all-atom molecular dynamics (MD) simulation. (**F**) Representative simulation snapshots of the membrane-binding motifs embedded in the *E. coli* membrane. Proteins are displayed in purple, and lipid tails are shown in cyan. Nitrogen and phosphorus atoms of the lipid head groups are represented in the van der Waals form in blue and gray, respectively. See *Supplementary file 6* for data statistics.

The online version of this article includes the following figure supplement(s) for figure 4:

**Figure supplement 1.** All-atom molecular dynamics (MD) simulations of membrane-targeting sequence (MTS) and LacY variants.

## Diffusion and localization of the MTS and TM segments

Unlike RNE in *E. coli*, RNase Y, a functional homolog of RNE in *B. subtilis*, is localized to the membrane via a TM domain (*Lehnik-Habrink et al., 2011*). We wondered if there are differences between a peripheral motif (like the MTS of *E. coli*'s RNE) and a TM motif in terms of membrane localization and mobility. To address this question, we created RNE mutants in which the MTS was replaced with a TM domain. For the TM domain, we used TM segments of LacY, a native *E. coli* protein, rather than using the TM motif from *B. subtilis* RNase Y, which might interact with the *E. coli* membrane in a non-native manner.

Before creating the RNE mutants, we characterized the MB% and the diffusion of individual short membrane-binding motifs. Native LacY contains 12 TM segments, arranged into two groups of six (*Abramson et al., 2003*). We successfully expressed constructs containing the first two TM segments (LacY (1–74) or LacY2), the first six TM segments (LacY (1–193) or LacY6), and the full-length LacY (LacY12), each fused to mEos3.2 and expressed from a chromosomal IPTG-inducible promoter (*Figure 4A*). We then imaged their membrane localization and diffusion. Both the MTS segment and LacY-derived TM segments showed a strong membrane enrichment (*Figure 4B*). In terms of diffusion, LacY2 and LacY6 diffused faster than the MTS segment (*Figure 4C*), contrary to expectations based on size (or mass)-dependent diffusion (*Figure 4D*).

According to the Stokes–Einstein relation for diffusion in simple fluids (*Miller and Walker, 1924*) and the Saffman–Delbrück diffusion model for membrane proteins (*Saffman and Delbrück, 1975*), $D$ decreases as particle size increases, albeit with different scaling behaviors. Specifically, the Saffman–Delbrück model predicts that $D$ for membrane proteins decreases logarithmically with increasing radius of the membrane-embedded region, assuming a constant membrane environment (*Saffman and Delbrück, 1975*). Thus, if size (or mass) were the primary determinant of diffusion, LacY2 and LacY6 would diffuse more slowly than the smaller MTS. The observed discrepancy instead implies that $D$ may be governed by how each motif interacts with the membrane. For example, the way that TM domains are anchored to the membrane may facilitate faster lateral diffusion with surrounding lipids.

Despite the prevalence of peripheral membrane proteins (*Papanastasiou et al., 2013*), how they interact with the membrane and how this differs from TM proteins remains poorly understood. To further explore this, we conducted all-atom molecular dynamics (MD) simulations of the MTS and the LacY variants interacting with the *E. coli* membrane using the NAMD software (*Phillips et al., 2005*). In the simulations, protein motion was calculated for 1 µs. Although the absolute $D$ values were higher than experimental values (possibly due to the absence of mEos3.2 in the model), the overall trends were preserved; among the LacY series, larger constructs diffused more slowly (*Figure 4E, F*, *Figure 4—figure supplement 1*). Most importantly, the MTS again diffused more slowly than LacY2 in silico (*Figure 4E*, *Figure 4—figure supplement 1*). By calculating membrane–protein interaction energies, we found that the MTS–membrane interactions were more stable than those of LacY2 (*Figure 4—figure supplement 1*). These results suggest that the slower diffusion of the MTS is due to stronger interactions with lipid head groups compared to membrane-embedded TM segments.

## RNE mutants carrying a TM motif

Since LacY2 and LacY6 showed strong membrane enrichment similar to LacY12 (*Figure 4B*), we replaced the MTS in RNE with LacY2, LacY6, and LacY12 in the presence or absence of the CTD (*Figure 5A, B*, *Figure 5—figure supplement 1*). All chimeric RNE mutants were expressed from the native chromosomal locus as the only copy of *rne*, with mEos3.2 fused at the C terminus for imaging. The resulting strains exhibited no noticeable differences in growth rate compared to the WT strain (*Supplementary file 3*), suggesting that the RNE mutants were functionally active.

xNorm histograms of the ΔCTD mutants indicated membrane localization similar to LacY (*Figure 5C*). However, mutants containing the CTD showed noticeable cytoplasmic subpopulations when LacY2 and LacY6 were used in place of the MTS (*Figure 5D*, *Appendix 1—figures 1 and 2*). Mathematical model fitting of the xNorm histograms estimated the MB% of RNE-LacY2-CTD and RNE-LacY6-CTD to be 69% [66%, 73%] and 86% [84%, 90%], respectively (*Figure 5F*). We note that imperfect membrane localization was observed only in mutants containing the CTD; the same protein without the CTD showed MB% of 100% (*Figure 5E, F*, *Appendix 1—figure 1*). These findings suggest that the CTD may contribute to unstable membrane binding of RNE. Supporting this idea, previously characterized RNE MTS point mutants with MB%<50% (*Figure 2D, E*) also exhibited

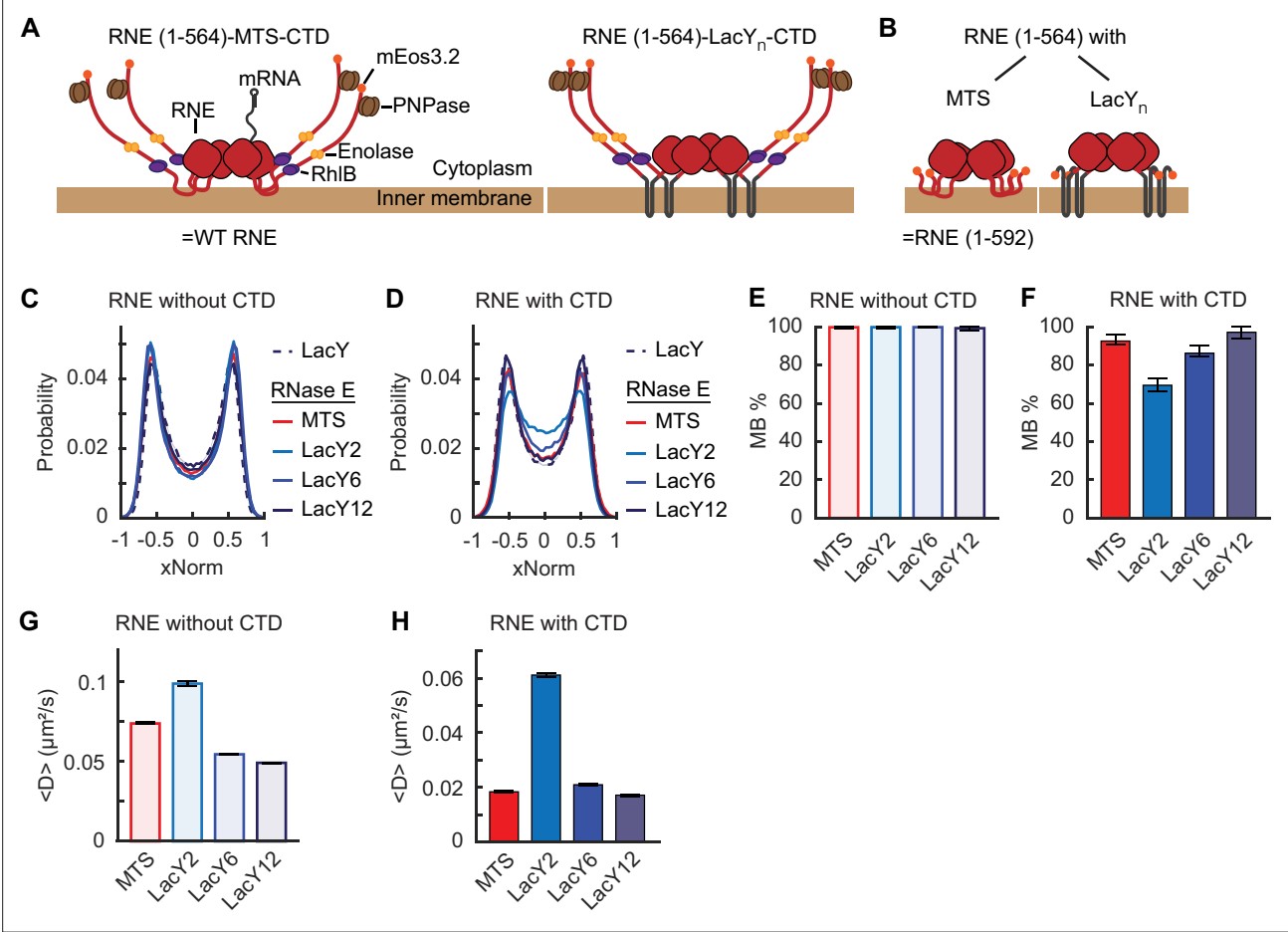

**Figure 5.** Localization and diffusion of chimeric RNE with or without the C-terminal domain (CTD). Cartoon schematic of RNE chimeric variants with the CTD (**A**) and without the CTD (**B**). They are not to scale. (**C, D**) xNorm histograms of chimeric RNE localization compared with that of LacY. The SEM from bootstrapping is displayed but smaller than the line width. Membrane-binding percentage (MB%) of chimeric RNE mutants without the CTD (**E**) or with the CTD (**F**) with various membrane-binding motifs. Error bars are from a 95% confidence interval. Mean diffusion coefficients of chimeric RNE without the CTD (**G**) or with the CTD (**H**). Error bars are the SEM. Each dataset contains at least 70,000 tracks for diffusion or 72,000 spots for xNorm. See *Supplementary file 6* for data statistics.

The online version of this article includes the following figure supplement(s) for figure 5:

**Figure supplement 1.** Linear representation of RNE monomer in various mutants used in this study.

**Figure supplement 2.** Clustering of RNE in *E. coli*.

increased MB% upon the CTD removal (*Appendix 1—figure 1*). Such a difference in MB% between the CTD-containing and the CTD-lacking mutants was not observed in the chimera based on LacY12 (*Figure 5E, F*), possibly due to the stable membrane insertion by LacY12.

Next, we examined whether the *D* of chimeric RNE mutants varied depending on the type of membrane-binding motifs. For example, the MTS diffused more slowly than LacY2 and LacY6, despite being smaller in size (*Figure 4C, D*). Based on this, we expected the chimeric RNE with LacY2 or LacY6 to diffuse faster than RNE with MTS. Indeed, in the absence of the CTD, we found that the *D* of LacY2-based RNE was 1.33 ± 0.01 times as fast as the MTS-based RNE (*Figure 5G*). However, LacY6-based RNE did not diffuse faster than the MTS-based version (*Figure 5G*). This result may be due to the high TM load (24 helices) created by four LacY6 anchors in the RNE tetramer. Although all constructs are tetrameric, the 24-helix load (LacY6), compared with 8 (LacY2) and 4 (MTS), likely enlarges the membrane-embedded footprint and increases drag, thereby changing the mobility advantages assessed as standalone membrane anchors.

In the presence of the CTD, the *D* of LacY2 and LacY6-based RNE became 3.33 ± 0.004 and 1.14 ± 0.01 times that of the MTS-based RNE counterpart (i.e., the WT RNE), respectively (*Figure 5H*). This

is likely influenced by the presence of a cytoplasmic population (~31% for LacY2 and ~14% for LacY6; *Figure 5F*, *Appendix 1—figure 2J*), which diffuses more rapidly than membrane-bound molecules (possibly by a factor of five, based on *Figure 3B*). Taken together, our data suggest that the CTD weakens the membrane association of RNE and small TM motifs can facilitate the diffusion of RNE.

## Functional consequence of subcellular localization and diffusion of RNE

To check the functional consequence of cytoplasmic localization of RNE, we measured *lacZ* mRNA degradation in various RNE mutants presented in this study. Recently, we developed an assay to quantify both co-transcriptional and post-transcriptional degradation rates of *lacZ* mRNA by inducing its transcription for only 75 s, thereby capturing the degradation of nascent mRNA (*Kim et al., 2024*). In WT cells, we found that the co-transcriptional degradation rate ($k_{d1}$) is about 10 times slower than the post-transcriptional degradation rate ($k_{d2}$) (*Kim et al., 2024*). In cells expressing RNE ΔMTS, however, $k_{d1}$ increases by a factor of ~3, suggesting that cytoplasmic RNE can freely diffuse and degrade nascent mRNAs (*Kim et al., 2024*). This result led us to hypothesize that RNE variants exhibiting a cytoplasmic subpopulation (*Figures 2E and 5F*) may exhibit a larger $k_{d1}$ compared to more membrane-bound variants.

We repeated this assay in a strain expressing the WT RNE fused to mEos3.2 (*Figure 6—figure supplement 1*). Note that this strain is different from the one we used for imaging because a monocistronic *lacZ* gene is needed for the transient induction assay (*Kim et al., 2024*). The relative abundances of 5' *lacZ* mRNA (Z5) remained constant prior to the rise in 3' *lacZ* mRNA (Z3) levels (between ~100 and 210 s), confirming negligible co-transcriptional degradation in WT cells (*Kim et al., 2024*; *Figure 6A*). However, in cells expressing RNE-LacY2-CTD (MB% = 69%), Z5 levels exhibited a downward trend during the same time window (*Figure 6B*). The estimated $k_{d1}$ was close to what was observed in RNE ΔMTS (*Kim et al., 2024*) (gray line, p = 0.28; *Figure 6C*). A similarly high $k_{d1}$ was also observed in RNE MTS point mutations, F582E and F757E, which exhibited a ΔMTS-like xNorm profile (*Figure 2D*), supporting the idea that the cytoplasmic subpopulation of RNE enables co-transcriptional mRNA degradation in *E. coli* (*Figure 6C*). Notably, RNE-LacY6-CTD, which also exhibited a cytoplasmic subpopulation (~14% from MB% of 86%), did not exhibit a significant increase in $k_{d1}$ (*Figure 6C*), suggesting a critical amount of cytoplasmic population may be needed to facilitate co-transcriptional mRNA degradation.

We next examined whether the post-transcriptional mRNA degradation rate ($k_{d2}$) is limited by the slow diffusion of membrane-bound RNE. In the presence of the CTD, $k_{d2}$ did not significantly vary across membrane-targeting variants (*Figure 6D*). For example, even though the LacY12 motif slows RNE diffusion, its $k_{d2}$ was similar to that of WT RNE (*Figure 6D*). A critical control for this comparison is RNE abundance: because RNE autoregulates its expression by degrading its own transcript (*Schuck et al., 2009*), slower diffusion could elevate RNE levels and mask the negative effect of the reduced mobility. We tested this explicitly and found that overexpression of WT RNE does not significantly change $k_{d1}$ or $k_{d2}$ (*Figure 6—figure supplement 2*). Thus, when the CTD is present, neither copy number nor diffusion of membrane-bound RNE limits $k_{d2}$.

In the absence of the CTD, $k_{d1}$ was high in RNE variant F582E (MB% = 67%; *Appendix 1—figure 1*), reaching levels close to its cytoplasmic counterpart, the ΔMTS ΔCTD mutant (gray line, p = 0.7; *Figure 6E*). $k_{d2}$ values for ΔCTD variants were, in general, lower than those of their CTD-containing counterparts (*Figure 6F*), indicating the importance of the CTD for the catalytic activity. These findings are consistent with previous reports on RNE constructs based on the MTS (*Lopez et al., 1999*; *Leroy et al., 2002*; *Islam et al., 2021*; *Kim et al., 2024*). Among ΔCTD variants with different membrane motifs, point mutants F575E and F582E (MB% = 91% and 67%, respectively; *Appendix 1—figure 1*) exhibited higher $k_{d2}$ than the MTS-based variant, in agreement with the fact that completely cytoplasmic ΔMTS ΔCTD exhibited higher $k_{d2}$ than the MTS-containing ΔCTD (*Figure 6F*). However, the LacY2-based RNE variant, which diffuses faster than the MTS version (*Figure 5G*), did not show a corresponding increase in $k_{d2}$ (*Figure 6F*). Plus, LacY6 and LacY12 versions showed even lower $k_{d2}$ than MTS-based RNE ΔCTD. Overall, these results indicate that the reduced $k_{d2}$ caused by ΔCTD cannot be rescued by faster diffusion of membrane-bound RNE. In fact, it may be further impaired by large and slow membrane motifs (such as LacY12). The presence of the CTD appears to buffer the effects of large membrane-binding motifs on RNE's catalytic activity, helping to maintain efficient post-transcriptional mRNA degradation ($k_{d2}$).

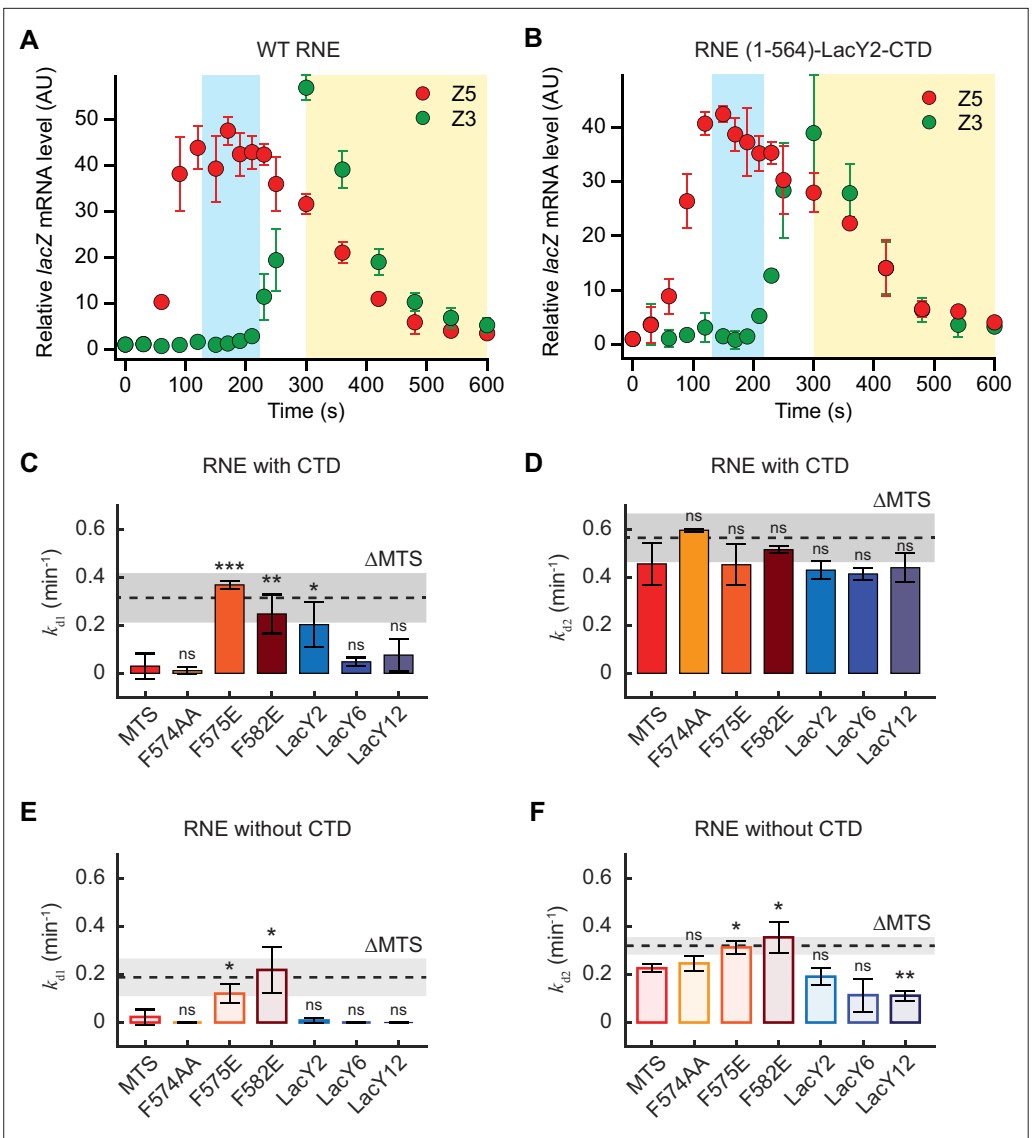

**Figure 6.** *lacZ* mRNA degradation rates in RNE mutant strains. *lacZ* mRNA levels in WT RNE (**A**, strain SK595) and in RNE-LacY2-CTD (**B**, strain SK505) when *lacZ* transcription was induced with 0.2 mM IPTG at $t = 0$ s and re-repressed with 500 mM glucose at $t = 75$ s. Blue and yellow regions indicate the time windows used to measure $k_{d1}$ and $k_{d2}$, respectively, by exponential fitting of 5' *lacZ* mRNA (Z5) in individual replicates. Co-transcriptional and post-transcriptional *lacZ* mRNA degradation rates, $k_{d1}$ (**C, E**) and $k_{d2}$ (**D, F**), respectively, in various RNE mutants containing different membrane-binding motifs, either with the CTD (solid bars, **C, D**) or ΔCTD (light bars, **E, F**). The dotted lines indicate the $k_{d1}$ and $k_{d2}$ values of cytoplasmic RNE ΔMTS (strain SK339 in **C, D**) (***Kim et al., 2024***) or RNE ΔMTS ΔCTD (strain SK370, in **E, F**). In all panels, error bars represent the standard deviations from two to three biological replicates. Two-sample *t*-tests were performed relative to the MTS case in each graph (see ***Supplementary file 7*** for the p-values). ***, ** and * indicate p<0.001, p<0.01 and p<0.05, respectively, and ns indicates a statistically nonsignificant difference.

The online version of this article includes the following figure supplement(s) for figure 6:

**Figure supplement 1.** *lacZ* mRNA degradation rates when RNE is fused to mEos3.2 or not (SK595 vs. SK98).

**Figure supplement 2.** Effect of RNE overexpression on *lacZ* mRNA degradation kinetics.

## Discussion

Our study establishes the membrane enrichment and slow diffusion of RNE in *E. coli*. This supports the notion that sequestration of RNE on the membrane confers the spatial and temporal separation between synthesis and decay of mRNAs (***Kim et al., 2024***). Furthermore, the processing of rRNA

(*Apirion, 1978*; *Bessarab et al., 1998*; *Ghora and Apirion, 1978*; *Li et al., 1999*; *Roy et al., 1983*) and tRNA (*Li and Deutscher, 2002*; *Ow and Kushner, 2002*) and small RNA-based gene regulations (*Ikeda et al., 2011*; *Reyer et al., 2021*) mediated by RNE likely take place on the membrane.

For WT RNE, our analysis showed MB% of 93%, close to 91% previously reported using immuno-gold labeling and freeze-fracture electron microscopy (*Liou et al., 2001*). Whether the MB% of RNE changes under different growth conditions remains to be tested. As a case study, we examined the MB% of WT RNE when cells were grown in M9 minimal medium with succinate as the sole carbon source without supplements, a condition different from that used in our primary experiments (M9 glycerol with supplements). The MTS segment exhibited a reduction in MB% from 100% to 83% (*Appendix 1—figure 1*), suggesting that MTS-mediated membrane binding can be sensitive to growth conditions. However, the MB% of WT RNE was only marginally affected by the same media change (*Appendix 1—figure 1*). We speculate that tetramer formation stabilizes RNE membrane localization, even when individual MTS motifs exhibit a weak membrane affinity. A similar phenomenon has been observed with MinD; although its amphipathic motif alone is cytoplasmic, MinD becomes membrane-associated upon dimerization, indicating that oligomerization enhances the membrane binding of this peripheral membrane protein (*Szeto et al., 2003*). By analogy, the tetrameric structure of RNE may reinforce membrane association, buffering against changes in cellular metabolism or lipid environment that would otherwise weaken MTS-mediated binding.

While individual TM motifs exhibited strong membrane association regardless of growth conditions (*Appendix 1—figure 1*), we found LacY2-based RNE can lose membrane-binding affinity, unlike MTS-based RNE. The loss of MB% in LacY2-based RNE was observed only in the presence of the CTD (*Appendix 1—figure 1*), suggesting that the CTD negatively affects membrane binding of RNE, possibly by altering protein conformation. In fact, all ΔCTD RNE mutants we tested exhibited higher MB% than their CTD-containing counterparts (*Appendix 1—figure 1*). For example, WT RNE (containing CTD) showed an MB% of 93%, whereas its ΔCTD version, RNE (1–592), showed 100%. Similarly, the chimeric RNE-LacY6-CTD showed an MB% of 86%, while its ΔCTD version showed 100% (*Figure 5E, F*; *Appendix 1—figure 1*). A similar trend was observed for MTS point mutants (*Appendix 1—figure 1*), further supporting that the CTD decreases membrane association across RNE variants. We speculate that this effect may be related to the CTD's role in promoting phase-separated ribonucleoprotein condensates, as observed in *Caulobacter crescentus* (*Al-Husini et al., 2018*). In *E. coli*, we also observed a modest increase in the clustering tendency of RNE compared to ΔCTD (*Figure 5—figure supplement 2*).

The $D_{\text{RNE}}$ we measured in *E. coli* (0.018 μm²/s) is comparable to those measured in other bacterial species. For example, in *C. crescentus*, the *D* of its cytoplasmic RNE was shown to be about 0.03 μm²/s (*Bayas et al., 2018*). The diffusion of RNase Y in *B. subtilis* was found in either a slow (0.031 μm²/s) or fast (0.3 μm²/s) population (*Oviedo-Bocanegra et al., 2021*), with the slow population corresponding to RNase Y bound to mRNA and/or the putative RNA degradosome and the fast population representing freely diffusing RNase Y. In the case of *E. coli* RNE, a two-population fit of the *D* histogram based on MB% identified the slow and fast subpopulations whose *D* values differed by a factor of 4 (*Figure 3—figure supplement 1*). This agrees with our finding that cytoplasmic RNE ΔMTS diffuses ~5 times as fast as WT RNE on average (*Figure 3B*).

Diffusion is affected by the size of the particle in a given medium (*Saffman and Delbrück, 1975*), and it has been used to identify different size forms of a protein due to biochemical interactions or complex formation (*Kapanidis et al., 2018*; *Sanamrad et al., 2014*). Related, RNA substrates interacting with RNE can also increase the effective mass of RNE and lower its *D* value. However, when cellular mRNAs were depleted by rifampicin treatment, RNE diffusion increased less than that of other RNA-binding proteins related to transcription and translation. For example, the large and small ribosomal subunits (*Bakshi et al., 2012*; *Gray et al., 2019*; *Sanamrad et al., 2014*), tRNA (*Volkov et al., 2018*), and RNA polymerase (*Stracy et al., 2015*) showed a large (about 10- to 20-fold) increase in *D* upon rifampicin treatment. Interestingly, Hfq, an RNA chaperone involved in small RNA regulation together with RNE, showed a moderate increase (~2-fold) in *D* when RNA was depleted by rifampicin (*Park et al., 2021*). We note that our result is consistent with previous studies that examined the effect of rifampicin on RNE diffusion in *E. coli* (*Strahl et al., 2015*; *Hamouche et al., 2021*) as well as RNE in *C. crescentus* (*Bayas et al., 2018*) and RNase Y in *B. subtilis* (*Hamouche et al., 2020*; *Oviedo-Bocanegra et al., 2021*). One possible explanation is that RNA-bound RNE (and RNase Y)

is short-lived compared to our frame interval (~20 ms), unlike other RNA-binding proteins related to transcription and translation, interacting with RNA for ~1 min for elongation (*Dai et al., 2016*).

Lastly, the slow diffusion of the MTS in comparison to LacY2 and LacY6 suggests that MTS is less favorable for rapid lateral motion in the membrane. Our MD simulations demonstrated that this reduced diffusivity may originate from a stronger interaction energy between the protein and the membrane. Whether this property is specific to the MTS or generalizable to other peripheral versus integral membrane motifs remains to be tested. We speculate that this can be a general phenomenon, as peripheral motifs interact with lipids orthogonally while integral membrane motifs align parallel to the bilayer and may diffuse more freely by coupling with the motion of surrounding lipids. The diffusion behavior of membrane-bound proteins reflects underlying protein–lipid interactions and membrane dynamics (*Knight et al., 2010*). Accordingly, future work may analyze how protein–lipid interaction strength and membrane dynamics contribute to differences in lateral diffusion between peripheral and integral membrane proteins.

Altogether, our work highlights strategies to modulate the MB% and diffusion of RNE to possibly affect mRNA degradation rates. For example, membrane-bound RNE lacking the CTD can stabilize mRNAs and increase protein expression. This idea has been used in a commercial *E. coli* BL21 strain (Invitrogen's One Shot BL21 Star) to increase recombinant protein yield (*Lopez et al., 1999*). Additionally, RNE variants with environmentally responsive MB%, such as RNE-LacY2-CTD, could be harnessed to tune mRNA half-lives and protein expression levels under different growth conditions.

# Materials and methods

## Bacterial strains

Strains used in this study are listed in *Supplementary file 1*, and details of strain construction are in *Supplementary file 2*. All the engineered regions were confirmed by DNA sequencing. Further information and requests for strains should be directed to the corresponding author.

Doubling times were measured from optical density at 600 nm ($OD_{600}$) using a microplate reader (Synergy HTX multi-mode reader, BioTek). Cultures were first grown overnight at 30°C and diluted $10^3$- to $10^4$-fold in the same type of fresh media. Cultures were pipetted into a 96-well plate and grown for 24 hr at 30°C with continuous shaking and $OD_{600}$ measurements every 3 min.

The strains grown in M9 succinate were similarly grown overnight at 30°C but then diluted 100-fold in the same type of fresh media before growing for an additional ~24 hr. Cells were then diluted another 100-fold and pipetted into a 96-well plate for growth monitoring over 48 hr in the plate reader.

For data analysis, the background reading was removed by subtracting the initial $OD_{600}$ value, and the resulting values between 0.001 and 0.1 were fit to an exponential function to determine doubling time (listed in *Supplementary file 3*). The fluorescent protein fusion had no measurable effect on the doubling time compared with WT cells.

## Sample preparation and cell growth for single-molecule imaging

Glass slides (Fisher 12-544-1) and #1.5 coverslips (Fisher 12544A or VWR 16004-344) were cleaned by three rounds of sonication in 100% ethanol, 70% ethanol, and Milli-Q water, respectively. They were stored in Milli-Q water until use and dried by nitrogen gas right before usage.

1% wt/vol agarose pads were created by melting agarose (Invitrogen 16500–100) in fresh liquid medium used for cell growth.

Unless otherwise stated, we grew cells in M9 minimal media supplemented with 0.2% vol/vol glycerol (Invitrogen 15514011), 0.1% wt/vol casamino acids (Bacto 223050), and 1 μg/ml thiamine (Research Products International T21020) at 30°C. A few colonies were inoculated in the M9 medium for at least 6 hr at 30°C with shaking at 220 rpm. This starter culture was diluted 3000-fold or more for overnight growth. When the $OD_{600}$ reached 0.150–0.250, cells were placed on an agarose pad prepared on a glass slide. Excess liquid was removed by air drying for about a minute and then a cleaned coverslip was laid on the top and sealed with VALAP. The sample was imaged immediately. LacY, LacY2, LacY6, MTS, and RNE with *lacZ* overexpressed cells were induced with a final concentration of 1 mM IPTG for at least 90 min or up to 16 hr before imaging.

## Drug treatment

To see the effect of mRNA amount on the diffusion of proteins of interest, cells were treated with either rifampicin (400 µg/ml) for 15 min or chloramphenicol (100 µg/ml) for 30 min in the liquid culture with shaking at the growth temperature. Corresponding agarose pads for imaging included either 200 µg/ml rifampicin or 100 µg/ml chloramphenicol. We added these compounds when the molten agarose was at 55–65°C.

## Single-molecule imaging

We took bright-field images and fluorescence images using a Nikon Ti-2 microscope equipped with a TIRF objective 100x/1.49 NA (Nikon), a 4-color laser launch (iChrome MLE-LFA-NI1), and an EM CCD camera (Andor iXon Ultra). For single-particle tracking photoactivated localization microscopy (sptPALM) (*Manley et al., 2010*), fluorescent images were taken using two lasers. A 405-nm laser was used to stochastically convert mEos3.2 from the green to the red fluorescence structure with a power between 0.003 and 0.055 W/cm$^2$ depending on the number of molecules of interest. A 561-nm laser was used to excite the photo-converted mEos3.2 molecules with a power between 2.3 and 2.7 W/cm$^2$. Images were taken with a frame rate of 21.7 ms with continuous laser illumination for 3 min. Each cell strain was imaged on at least three separate days, and 6–16 movies were collected each day.

All images were acquired using the Nikon Elements software (Nikon). We used a quasi-TIRF laser angle to reduce background. The same angle was used for all imaging.

## Nucleoid imaging

We imaged the nucleoid region of cells containing RNE-mEos3.2 and HU-mCherry (SK512) grown in the same way as our single-molecule imaging samples. We used a Nikon Ti-2 microscope equipped with a phase-contrast Plan Apochromat objective 100x/1.45 NA (Nikon), a Sola SE II 365 light engine (Lumencor), an ET/mCh/TR filter cube (Nikon, 96365), and an Orca-R2 CCD camera (Hamamatsu Photonics). All images were acquired using the Nikon Elements software. Cells were treated with chloramphenicol in the same way as in single-molecule imaging of RNE-mEos3.2 (SK187). We also imaged cells without antibiotic treatment as a control.

## mRNA degradation rate measurement

We measured mRNA levels and degradation rates as described previously (*Kim et al., 2024*). Briefly, cells were grown the same way as for imaging experiments. When cells were in the early exponential phase (OD$_{600}$ ~0.02), cells were induced with 0.2 mM IPTG. The IPTG addition marks time zero ($t$ = 0). At $t$ = 75 s, cells were re-repressed with 500 mM glucose. During this time course, cells were sampled every 20–60 s for total RNA extraction using PureLink RNA Mini Kit (Life Technologies). Next, *lacZ* mRNA levels were quantified by qRT-PCR using KAPA SYBR FAST qPCR Master Mix (KAPA Biosystems) and CFX Connect Real-Time System (Bio-Rad). Primer sequences for 5′ end and 3′ end mRNA regions are provided in *Supplementary file 5*.

For the RNE overexpression strain (SK394) shown in *Figure 6—figure supplement 2*, cells were inoculated in the same way as other strains, and 0.2% arabinose was added at the time of dilution for overnight growth. When OD$_{600}$ became 0.1 the next day, cells were washed with fresh media of the same volume (with no arabinose), and the culture was shaken until OD$_{600}$ reached 0.2.

## Cell fixation

To fix *E. coli* cells, we followed the protocol we have established previously (*Kim and Vaidya, 2020*). Namely, cells were grown under the same conditions as used for microscopy. Once they reached the exponential phase, cells were fixed with 4% formaldehyde in 0.03 M disodium phosphate buffer (pH 7.4) for 15 min at room temperature, followed by an additional 30 min on ice. After fixation, cells were washed three times with PBS. For each wash, cells were pelleted by centrifugation at 5000 × *g* for 4 min at room temperature. Cells were then resuspended in PBS and kept on ice until imaging. All samples were imaged on the same day as fixation.

## Single-molecule tracking data analysis

Data was analyzed in Matlab using Oufti (*Paintdakhi et al., 2016*) for cell detection from bright-field images, u-track (*Jaqaman et al., 2008*) for fluorescent spot detection and linking into tracks, and a

custom-built Matlab code called spotNorm for finding a fluorescent spot's normalized position in a cell (*Figure 1A–C*). After this, the diffusion and subcellular localization were analyzed as described below.

## Oufti

Cell outlines (or cell mesh) were computed from bright-field images using Oufti software developed by Jacobs-Wagner Lab (*Paintdakhi et al., 2016*). The following parameters were used:

    edge mode for detection = valley, with dilate = 2
    openNum = 1
    InvertImage = 0
    ThreshFactorM = 0.63493
    ThreshMinLevel = 0.795
    EdgeSigmaL = 1
    EdgeSigmaV = 7
    ValleyThresh1=0.14286
    ValleyThresh2=0.77143

Only cells that are isolated (not touching other cells) and not dividing (based on cell constriction) were used in the analysis.

## u-track

Fluorescent spots were detected and compiled into tracks using u-track software developed by Danuser Lab and Jaqaman Lab (*Jaqaman et al., 2008*). The parameters for fluorescent spot detection were as follows: pixel size of 160 nm, time interval of 0.0217 s, numerical aperture of 1.49, and camera bit depth of 16 bits. Spot detection was done via Gaussian mixture-model fitting with the Gaussian standard deviation set to 0.9 pixels and the alpha value for comparison with local background was set to 0.015. Also, the hypothesis test alpha value for Gaussian fitting at local maxima was set to an amplitude of 0.015 and distance of 0.05. For the tracking step, we used the default settings for 2D particle tracking with the maximum gap to close changed to 0 to avoid gaps in a track. Additionally, the minimum length of track segments from the first step was set to 4 frames. This setting ensures that all trajectories from u-track are at least frames long.

A notable parameter in u-track is the max pixel linking distance. We set it to 1–3 pixels depending on the preliminary diffusion coefficient based on trajectories identified using u-track's default 5-pixel maximum linking distance, $D_{5pix}$. We found that the max linking distance of 5 pixels identifies artificial large random jumps in our dataset that falsely creates a large diffusion coefficient. To avoid this error, we set the max linking distance radius as ceiling($2*sqrt(4D_{5pix}\Delta t)$), where $\Delta t$ is the frame time (0.0217 s). Alternatively, we can use 1 pixel if $D_{5pix} < 0.0737$ μm$^2$/s and 2 pixels if 0.0737 μm$^2$/s $< D_{5pix} < 0.295$ μm$^2$/s.

## spotNorm

Fluorescent spot positions in (*x,y*) pixel coordinates, obtained from u-track, were transformed into normalized coordinates along the cell's long axis (yNorm) and short axis (xNorm) using a custom-built MATLAB script called spotNorm. First, the fluorescent spot position (in pixels) was compared to the cell outlines (in pixels) from Oufti of all cells for the corresponding fluorescent movie. Only spots within a cell were used in the downstream analysis. The center of the cell outline and the centroid region of spots inside of the cell were compared to check for drift between taking the bright-field image and the start of fluorescence tracking. If the centroid regions were different, then the average distance between the cell outline and the fluorescent data of all the cells in the frame was used to calculate a correction shift for fluorescent data. The spot's position was then projected onto the long and short axes of the cell, and the projected distances were normalized by the corresponding cell axis lengths. A positive or negative value for the spot's xNorm position was determined by the sign of the cross-product of the spot within the center line of the cell. xNorm values were between –1 and 1. yNorm values were between 0 and 1 where 0 and 1 are the two different cell poles.

By combining xNorm data from many cells, we created xNorm histograms. Spots included in these histograms were only from the cylindrical portion of the cell, not from the hemispherical endcaps. This

was calculated based on the spot position along the long axis, $y$, being within $R < y < (L - R)$, where R is half of the mean cell width and L is the length of the cell. For better statistics, we included spots from the first frames of a track in the xNorm histogram. Because we set a minimum track length of 4 frames in u-track analysis, each track contributes 4 points equally to the xNorm histogram. The bin size for xNorm histograms is 0.04. In the xNorm histogram, the error was calculated from the SEM from bootstrapping. Each xNorm histogram is displayed with SEM-based error ranges by a shaded area, but it is smaller than the data line width. RNE's xNorm histogram error shaded region is compared to error bar lines (*Figure 1—figure supplement 1*).

Diffusion: Tracks lasting at least 12 frames long were used for diffusion analysis. Time-averaged MSD was calculated according to *Equation 1*, and the diffusion coefficient (*D*) was calculated by a linear fit using *Equation 2* (*Savin and Doyle, 2005*):

$$MSD\left(\tau\right) = \frac{1}{N_\tau} \sum_{i=1}^{N_\tau} \left[\vec{r}\left(t_i\right) - \vec{r}\left(t_i + \tau\right)\right]^2 \tag{1}$$

$$MSD\left(\tau\right) = 4D\tau + b \text{ where } b = -\frac{4D\Delta t}{3} + 4\sigma^2 \tag{2}$$

In *Equation 1*, $\vec{r}\left(t_i\right)$ is the 2D vector location of a particle at time $t_i$, $\tau$ is the lag time, and $N_\tau$ is the number of frames that are averaged over in a track for that lag time. In *Equation 2*, $\Delta t$ is the exposure time, and $\sigma$ is the localization error. The y-intercept, $b$, is composed of two terms. The first term is a correction for dynamic error, and the second term is for static error (*Savin and Doyle, 2005*). We only fit the first three time points of MSD with *Equation 2* to get *D*. This ensures that the fit is done on short time delays (the first 25% of the minimum track length) to avoid MSD at large time intervals with low statistics (*Saxton, 1997*; *Wieser and Schütz, 2008*). Once we obtained *D* from individual tracks, we calculated their mean and SEM (standard deviation divided by the square root of the sample size) to report mean ± SEM of the diffusion coefficient.

We checked slow and fast subpopulations of proteins by analyzing the distribution of *D* of individual tracks. First, we fit the distribution of $D_{RNE}$ using a single Gaussian distribution (*Equation 3*) or a two-population distribution (*Equation 4*; *Figure 3—figure supplement 1*). We assumed MB% for the slow population and the rest for the fast population in the two-population fitting (*Figure 3—figure supplement 1*). The residuals from a one-population fit and a two-population fit were similar for RNE, indicating that the single Gaussian fit can be used. The distribution of *D* of ribosomal proteins (L1) showed a long tail, suggesting more than one mobility population exists, in agreement with other studies reporting multiple mobility populations for ribosomal proteins (*Bakshi et al., 2012*; *Sanamrad et al., 2014*). Hence, we used a two-population Gaussian distribution (*Equation 4*) for fitting (*Figure 3—figure supplement 3*). The slow population is likely from polysomes, or the ribosomes bound to mRNA (as these form a larger complex), and the fast population may represent free subunits not engaged in translation,

$$g\left(D_h\right) = \frac{c}{\sigma\sqrt{2\pi}} \exp\left(-\frac{1}{2}\frac{\left(D_h - \mu\right)^2}{\sigma^2}\right), \tag{3}$$

$$g\left(D_h\right) = \frac{c}{\sqrt{2\pi}} \left[\frac{a}{\sigma_1}\exp\left(-\frac{1}{2}\frac{\left(D_h - \mu_1\right)^2}{\sigma_1^2}\right) + \frac{\left(1 - a\right)}{\sigma_2}\exp\left(-\frac{1}{2}\frac{\left(D_h - \mu_2\right)^2}{\sigma_2^2}\right)\right]. \tag{4}$$

In *Equation 3*, $c$ is the scaling factor, $D_h$ is the diffusion coefficients from *D* histograms, $\mu$ is the mean, and $\sigma$ is the standard deviation. Similarly, in *Equation 4*, $\mu_1$ and $\mu_2$ are the means of the first and second populations, $\sigma_1$ and $\sigma_2$ are the standard deviations of the first and second populations, $a$ is the fraction of the first population, and $(1 - a)$ is the fraction of the second population. As mean values are for diffusion coefficients, we also refer to $\mu$ as $D_1$ for the single Gaussian fitting and $\mu_1$ and $\mu_2$ as $D_1$ and $D_2$, respectively, for the two-population Gaussian fitting.

## Membrane-binding percentage

To model an xNorm histogram from a distribution of molecules in 3D space, we assume that a cell is a cylinder, where molecules are homogeneously distributed on the surface (membrane) or inside

(cytoplasm). We use a Cartesian coordinate, in which the long axis of the cylinder is aligned along the $y$-axis with its circular cross-section oriented vertically in the $x$–$z$ plane (**Figure 1—figure supplement 3**).

Since molecules are uniformly distributed, we mainly consider the distribution of molecules in the circular cross-section (perpendicular to the $y$-axis) and calculate a marginal distribution of the $x$-coordinates, as $P_{memb}$ and $P_{cyto}$ for molecules on the circumference and inside, respectively. Finally, the probability distribution is modeled as a mixture distribution of the molecules on the membrane and in the cytoplasm,

$$P(x) = m \cdot P_{memb}(x) + (1 - m) \cdot P_{cyto}(x), \tag{5}$$

where $m$ is the MB%.

In detail, the inner membrane is described as a circle with radius $r$. Define $z(x) = \sqrt{r^2 - x^2}$. Then, the conditional probability distributions for molecules on the membrane and in the cytoplasm are as follows:

$$dP^0_{memb} = \frac{dx}{\pi z}, \tag{6}$$

$$dP^0_{cyto} = \frac{2z \cdot dx}{\pi r^2}. \tag{7}$$

Next, we introduce imaging effects to the model as follows: the localization error was added by convolving the raw distribution with a 2D-Gaussian blur $g(x, z) = \phi(x, 0, \sigma) \cdot \phi(z, 0, \sigma)$, where $\phi(u, c, s) = e^{-(u-c)^2/(2 \cdot s^2)}/\sqrt{2\pi}s$ is a Gaussian kernel. Here, we assumed the same localization uncertainty ($\sigma$) in the $x$ and $z$ dimensions. While we could not justify this assumption, we found that the choice of uncertainty in the $z$ dimension minimally affects the final outcome (xNorm histogram), and hence the uncertainty in the $x$ dimension was used to simplify the model. Additionally, the imaging system has a certain depth of focus, beyond which the signal cannot be captured by the camera. We let the defocused range at the bottom part of the cell be given by $z < -f$ (**Figure 1—figure supplement 3**). With these adjustments, the molecular density along the $z$-axis is proportional to $\int_{-\infty}^{f} \phi(u, -z, \sigma) \, du = \Phi((f + z)/\sigma)$, where $\Phi$ is the cumulative distribution function of a standard Gaussian. Therefore, the conditional probability distributions are formulated as

$$dP_{memb} \propto \left( \Phi\left(\frac{f+z}{\sigma}\right) + \Phi\left(\frac{f-z}{\sigma}\right) \right) \cdot dP^0_{memb} * g, \tag{8}$$

$$dP_{cyto} \propto \left( \int_{-z}^{z} \Phi\left(\frac{f+z}{\sigma}\right) du \right) \cdot \frac{dP^0_{cyto}}{z} * g. \tag{9}$$

Here, $\propto$ simply means the density should be scaled by normalizing constants.

To fit the experimental data to the model, parameters need to be scaled. The inner membrane radius $r$ is $R/dilF$, where $R$ is half of the cell width (experimentally obtained from bright-field images and Oufti analysis) and $dilF$ is the dilation factor, which accounts for the fact that membrane-bound molecules (from mEos3.2 imaging) are localized away from the cell boundary identified by the bright-field images (e.g., **Figure 1C**). In the end, $R$ is normalized to 1 to provide xNorm. Also, $f = R - fCut$ (**Figure 1—figure supplement 3**), and hence, $fCut$ is zero when the view of the cell is unobstructed. $\sigma = locErr/1000$, the localization error measured in μm. Theoretical xNorm distributions of membrane and cytoplasmic molecules are illustrated in **Figure 1—figure supplement 3**.

To estimate the MB% and other parameters in experimental xNorm data, we used the Markov-Chain Monte Carlo algorithm, which is an effective simulation approach for estimating Bayesian posterior distributions of multi-parameter models and their representative statistics. For better statistics, we fitted the absolute xNorm histogram. Specifically, we constructed a histogram by binning the experimental data (absolute xNorm from 0 to 1) and minimizing the mean-squared error to the model-implied values. All parameters ($dilF$, $locErr$, $fCut$, $m$) were free to change within the intuitive bounds (e.g., $m \in [0, 1]$), and they have uniform priors. We implemented the fitting procedure in Python using the *pymcmcstat* package (**Miles, 2019**). For robustness, we examined different choices of hyper-parameters (bin size, bounds, sampling numbers, and the burn-in period) and checked that the end result is insensitive to them. The fitting results are provided in **Supplementary file 4**.

## mRNA degradation rate

We estimated mRNA degradation rates by fitting a single-exponential decay model to the Z5 time-course data, as described previously (*Kim et al., 2024*). Briefly, the exponential fits were performed between $t$ = ~150 and ~210 s for $k_{d1}$ and between $t$ = ~300 and 600 s for $k_{d2}$.

## Statistical test

We tested for statistical similarity using p-values determined by a two-tailed Student's *t*-test on small datasets for *lacZ* mRNA degradation (*Supplementary file 7*). The two-tailed Student's *t*-test was calculated using the ttest2 function in MATLAB at the 5% significance level. When the alternative hypothesis was checked for differences, no additional specifications were used. When the alternative hypothesis was for the mean to be less than the other, we used the additional specification 'tail', 'left'. If the p-value is above 0.05, the two datasets are statistically similar. Otherwise, the alternative hypothesis is supported.

## Minimum diffusion coefficient that can be measured using our microscope

To test the minimum $D$ that our microscope can measure for mEos3.2, we prepared surface-immobilized mEos3.2 and measured $D$ of the stationary molecules. We purified His6-streptavidin-mEos3.2 expressed from a plasmid (SK567) in BL21 cells using His-Spin Protein Miniprep (Zymo Research, P2001) and then buffer exchanged into PBS (pH 7.4) using a centrifugal filter unit (Merck Millipore, UFC510024). His6-streptavidin-mEos3.2 was pipetted onto a coverslip functionalized with biotin and allowed to bind for 5 min before washing away excess liquid. After washing with Milli-Q water, it was sealed with a cleaned glass slide. Surface-immobilized mEos3.2 molecules were imaged the same way we imaged mEos3.2 in *E. coli*. The diffusion of His6-streptavidin-mEos3.2 immobilized on biotin coverslips was 0.0020 ± 0.0001 µm²/s (*Figure 3—figure supplement 1A*). This determines to what extent experimental noise between the fluorophore and microscope's localization error can cause apparent diffusion of a stationary molecule. This $D$ value was well below our measured diffusion coefficients of the molecules in this study, indicating slow diffusion coefficients are not due to tracking limitations.

## Detection efficiency of cytoplasmic versus membrane-bound proteins

Since proteins diffuse more rapidly in the cytoplasm than in the membrane, they might not be detected as well as the membrane-bound proteins in our microscopy experiments. To assess this potential undercounting, we compared the number of tracks in live versus fixed cells for WT RNE and for cyto-plasmic RNE ΔMTS. Fixed cells were imaged as previously described, except that the agarose pad was made of PBS instead of growth media. U-track was used to find the number of tracks in the cell using a max linking distance of 1 pixel. The mean number of tracks per cell in live and fixed cells was compared using the Fisher test (*Figure 1—figure supplement 2*). The Fisher-Irwin test is a statistical significance test used for contingency tables. The returned p-value was 0.18, indicating insignificant bias introduced by localization (and mobility) in detection.

## Cluster analysis

We analyzed clustering of RNE based on mEos3.2 images acquired from fixed cells, in which clustering is preserved (*Figure 5—figure supplement 2*). We used spatial point statistics used in a previous study of RNE clustering in *C. crescentus* (*Bayas et al., 2018*). For the analysis, we first chose cells with at least 300 spots (or tracks) located throughout the cell. A complete spatial randomness (CSR) was simulated as molecules homogeneously distributed on the membrane of a 3D cell using the same length, width, and number of points as each imaged cell. The simulated spots were projected to a 2D plane for comparison with experimental data.

Global clustering was quantified using Ripley's *K* function (*Ripley, 1977*) and its normalized forms (*Besag, 1977*). Ripley's *K* function represents the average probability of finding localizations at a distance *r* from the typical localization (*Kiskowski et al., 2009*). For CSR, $K(r)$ scales with the cell volume and is described by *Equation 10*, where $b_d r^d$ is the volume of the unit sphere in d dimensions. Deviations from this shape indicate clustering or repulsion. The normalized *K*-function is given by

$L(r)$ in *Equation 11*, which can further be normalized to the $H(r)$ function (*Ehrlich et al., 2004*). For example, the expected value of $K(r)$ for a random Poisson distribution is $\pi \cdot r^2$, and $L(r) = r$, $H(r) = 0$.

To analyze clustering across many cells, $\int_r H_{data}(r) - H_{CSR}(r)$ was calculated for each cell,

$$K(r) = b_d r^d, \tag{10}$$

$$L(r) = \sqrt[d]{K(r)/b_d}, \tag{11}$$

$$H(r) = L(r) - r. \tag{12}$$

## All-atom MD simulations

The all-atom simulations were performed using the NAMD 3.0 software (*Phillips et al., 2005*). The CHARMM-GUI Membrane Builder (*Jo et al., 2008*; *Jorgensen et al., 1983*; *Wu et al., 2014*) was utilized to generate *E. coli* lipid bilayers consisting of 80 POPE and 20 POPG lipids per leaflet. The lipid membranes were hydrated using the TIP3P water model. The protein–membrane system was equilibrated with the CHARMM36 force field (*Klauda et al., 2010*). Simulations were conducted under the NPT ensemble, maintaining the temperature at 310 K and pressure at 1 bar using the Nosé-Hoover Langevin piston and Langevin dynamics. A 2-fs time step was employed for integrating the equations of motion. van der Waals and electrostatic interactions were truncated at a 12-Å cutoff, with a switching function applied between 10 and 12 Å. The Particle-Mesh Ewald (PME) method was used to handle long-range electrostatic interactions. For MTS, we used the 17 amino acid sequence: QPGLLSRFFGALKALFS. For LacY2, we used the 74 amino acid sequence: MYYLKNTNFWMFGLFF FFYFFIMGAYFPFFPIWLHDINHISK SDTGIIFAAISLFSLLFQPLFGLLSDKLGLRK. For LacY6, we used the first 193 residues of LacY. For LacY12, we used the full-length LacY (417 residues).

We estimated diffusion by tracking the center-of-mass positions of the proteins over time. Ensemble-averaged MSD was calculated by averaging the squared displacements from 5 replicas (*Figure 4—figure supplement 1A*). The diffusion coefficient was then obtained based on the MSD = 4 Dt relationship.

To explain the unexpected diffusion dynamics of MTS relative to LacY2, we calculated the per-residue interaction energies between the protein and membrane systems using the NAMDEnergy plugin in VMD (*Figure 4—figure supplement 1B*). The interaction energy per residue for MTS is more negative compared to LacY2, indicating a stronger affinity for the lipid head groups. In contrast, LacY2 is primarily situated within the hydrophobic tails of the membrane lipids, showing a weaker interaction.

## Acknowledgements

We thank Drs. Mark Arbing, Agamemnon Carpousis, Johan Elf, and Christine Jacobs-Wagner for strains. We thank Maggie Liu, Kavya Vaidya, and Zach Wang for their contributions in the early phase of this work and the members of Kim lab for critical reading of the manuscript. This work was supported by the NSF Center for Physics of Living Cells (1430124), NSF Science and Technology Center for Quantitative Cell Biology (2243257), NIH (R35GM143203; R24GM145965), and Searle Scholars Program.

## Additional information

### Competing interests

Jeechul Woo: CEO of Moduli Technologies, LLC. Sangjin Kim: Reviewing editor, eLife. The other authors declare that no competing interests exist.

### Funding

| Funder | Grant reference number | Author |
|---|---|---|
| National Science Foundation | 1430124 | Sangjin Kim |

| Funder | Grant reference number | Author |
|---|---|---|
| National Science Foundation | 2243257 | Yu-Huan Wang<br>Emad Tajkhorshid<br>Sangjin Kim |
| National Institute of Health | R35GM143203 | Laura Troyer<br>Seunghyeon Kim<br>Brooke Ramsey<br>Sangjin Kim |
| National Institute of Health | R24GM145965 | Shobhna Shobhna<br>Emad Tajkhorshid |
| Searle Scholars Program | | Seunghyeon Kim<br>Sangjin Kim |

The funders had no role in study design, data collection, and interpretation, or the decision to submit the work for publication.

## Author contributions

Laura Troyer, Conceptualization, Resources, Data curation, Software, Formal analysis, Validation, Investigation, Visualization, Methodology, Writing – original draft, Writing – review and editing; Yu-Huan Wang, Resources, Data curation, Software, Formal analysis, Validation, Investigation, Visualization, Methodology, Writing – review and editing; Shobhna Shobhna, Formal analysis, Investigation, Visualization, Methodology, Writing – review and editing; Seunghyeon Kim, Resources, Formal analysis, Validation, Investigation, Methodology; Brooke Ramsey, Validation, Investigation, Visualization; Jeechul Woo, Resources, Software, Methodology; Emad Tajkhorshid, Supervision, Funding acquisition, Writing – review and editing; Sangjin Kim, Conceptualization, Resources, Supervision, Funding acquisition, Writing – original draft, Project administration, Writing – review and editing

## Author ORCIDs

Laura Troyer ⓘ https://orcid.org/0000-0002-7208-6654
Yu-Huan Wang ⓘ https://orcid.org/0009-0000-3086-1336
Seunghyeon Kim ⓘ https://orcid.org/0009-0000-5450-8860
Brooke Ramsey ⓘ https://orcid.org/0009-0004-1234-0042
Emad Tajkhorshid ⓘ https://orcid.org/0000-0001-8434-1010
Sangjin Kim ⓘ https://orcid.org/0000-0002-8444-7891

Reviewer #1 (Public review): https://doi.org/10.7554/eLife.105062.3.sa1
Reviewer #2 (Public review): https://doi.org/10.7554/eLife.105062.3.sa2
Reviewer #3 (Public review): https://doi.org/10.7554/eLife.105062.3.sa3
Author response https://doi.org/10.7554/eLife.105062.3.sa4

# Additional files

## Supplementary files

MDAR checklist

Supplementary file 1. List of strains used in this study.

Supplementary file 2. Strain construction.

Supplementary file 3. Doubling times and cell sizes.

Supplementary file 4. xNorm histogram modeling results.

Supplementary file 5. qRT PCR primers used in study.

Supplementary file 6. Figure data statistics.

Supplementary file 7. P-values determined by two-tailed Student's t-test.

Supplementary file 8. Diffusion coefficient (D) and 95% CI.

Source data 1. Metadata for microscopy data.

Source data 2. Protein localization data (xNorm, yNorm, and D).

Source data 3. Protein diffusion data (D of individual tracks).

## Data availability

Microscopy data generated in this study is provided as Source data 1–3. MATLAB code for data analysis and Python code for modeling are available at the publicly accessible site, https://github.com/sjkimlab/2025_RNaseE (copy archived in Zenodo: https://doi.org/10.5281/zenodo.17410099).

The following dataset was generated:

| Author(s) | Year | Dataset title | Dataset URL | Database and Identifier |
|---|---|---|---|---|
| Kim S | 2025 | sjkimlab/2025_RNaseE: v1.0 (v1.0) | https://doi.org/10.5281/zenodo.17410099 | Zenodo, 10.5281/zenodo.17410099 |

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

## Appendix 1

### Estimation of MB% from xNorm histograms

We analyzed xNorm histograms of mEos3.2-fused proteins using the fitting method described in Materials and methods. *Appendix 1—figure 1* shows all xNorm fitting results and *Supplementary file 4* lists the corresponding fitting parameters.

To further examine MB%, we plotted xNorm histograms after separating the localizations based on the mobility of the trajectories. Namely, we used a $D$ cutoff to examine xNorm of the fast and slow trajectories separately (*Appendix 1—figure 2*). Our hypothesis was that MB% of trajectories should be slow and their xNorm histograms should be LacY-like. To test this hypothesis, the $D_{cutoff}$ was chosen to match the fraction of the slow population ($D < D_{cutoff}$) with the MB% of that strain. Indeed, the xNorm for the slow populations typically had LacY-like xNorm profile, whereas the fast population had one flat central peak, similar to the distribution of RNE ΔMTS (*Appendix 1—figure 2*). Proteins that are either completely membrane-bound or cytoplasmic showed similar xNorm histograms for their slow and fast subpopulations. Examples include LacY, LacY6, RNE-LacY12-CTD, and RNE-F574AAΔCTD for completely membrane-bound, and LacZ, RNE ΔMTS, RNE ΔMTS ΔCTD, RNE-F582-CTD, and RNE-F575E-CTD for cytoplasmic. Note that only tracks with at least 12 frames will have a calculated $D$. Thus, only a subset of the xNorm data was used in the $D_{cutoff}$ xNorm histograms.

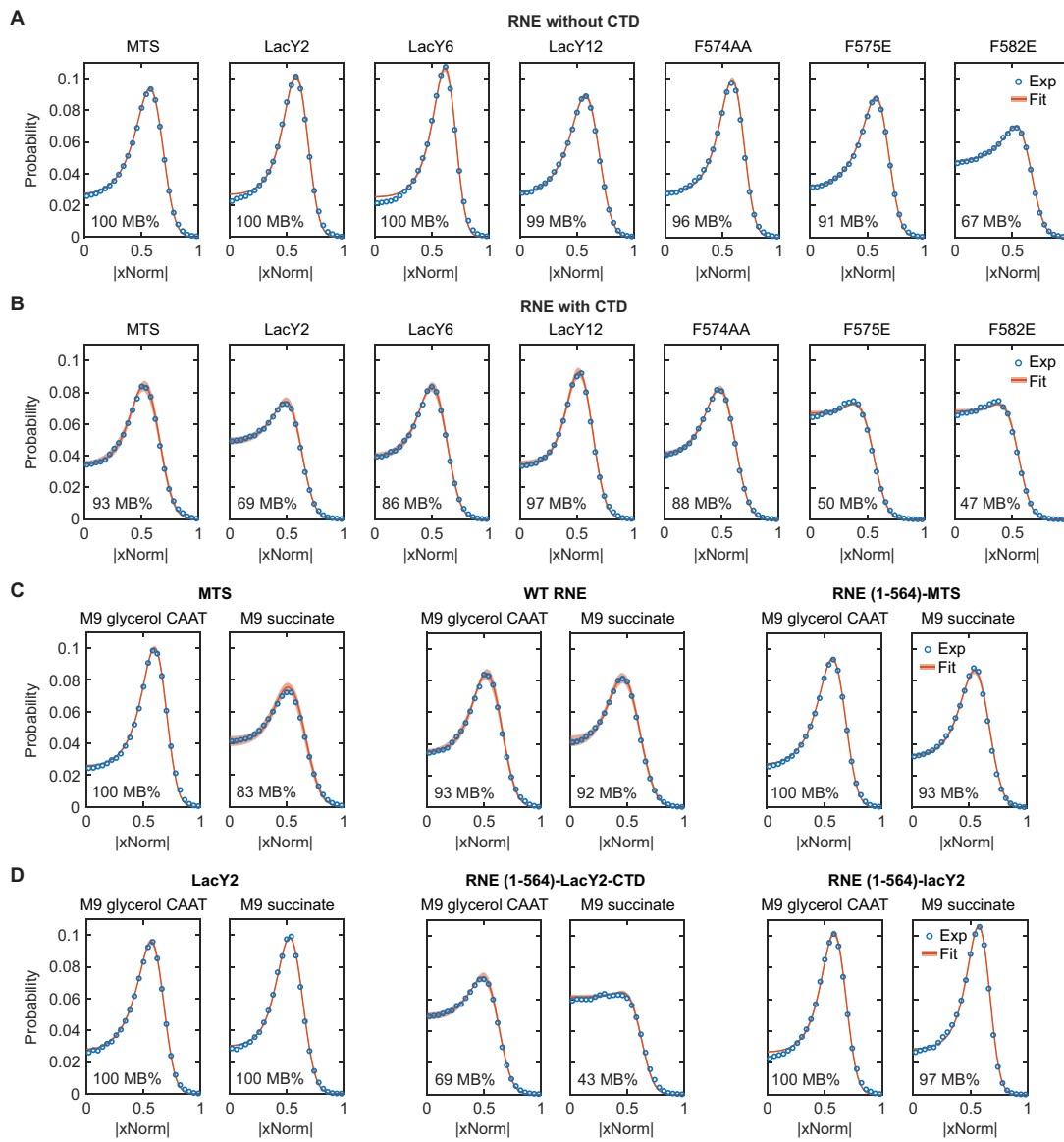

**Appendix 1—figure 1.** xNorm histogram of various RNE mutants used in this study. Chimeric RNE with various membrane-binding motifs without the CTD (**A**) or with the CTD (**B**). (**C**) Effect of growth media on MB% of the MTS motif, WT RNE, and RNE ΔCTD. (**D**) Effect of growth media on MB% of the LacY2 segment, RNE (1–564)-LacY2-CTD, and RNE (1–564)-LacY2. In all panels, experimental data (blue circles) is compared with MCMC-based fitting result (red). Red shaded regions indicate the expected xNorm histogram based on parameter values within the standard deviation. The MB% shown at the bottom left of each plot is the best-fitting result. See ***Supplementary file 4*** for xNorm histogram modeling results.

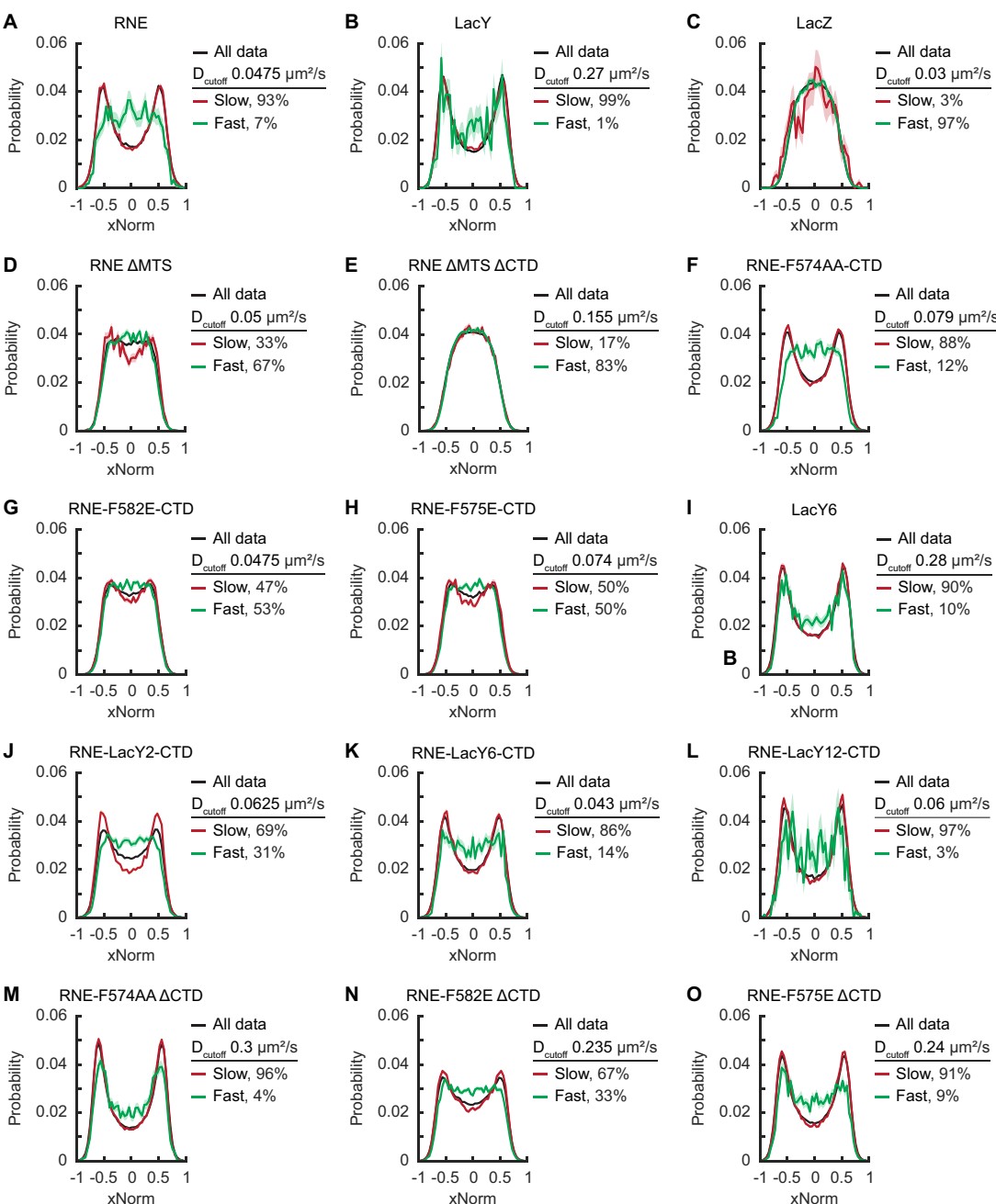

**Appendix 1—figure 2.** xNorm of protein constructs with MB% less than 99%. The xNorm histograms of protein constructs were analyzed by separating molecules (all data) into fast (green) and slow (red) subpopulations. These subpopulations were defined based on the diffusion coefficient $D$: molecules with $D$ values in the bottom MB% of the entire population (also below $D_{cutoff}$) were classified as slow, while the other molecules with $D$ values above the $D_{cutoff}$ were classified as fast. Protein constructs with uniform membrane or cytoplasmic localization, such as LacY and LacZ, respectively, showed similar xNorm histograms for both fast and slow populations (e.g., panels B, C, E, I, L, and M). See **Supplementary file 6** for data statistics.

## Appendix 2

## Calculation of the mass of the RNA degradosome

The canonical RNA degradosome in *E. coli* is comprised of RNE, enolase, PNPase, and RhlB (*Carpousis, 2007*; *Mackie, 2013*). However, there are other proteins that have been shown to associate with the degradosome, such as Ppk, CsdA, and PAP (*Blum et al., 1997*; *Carabetta et al., 2010*; *Jaso-Vera et al., 2021*).

We calculated the mass of the RNA degradosome assuming a fully loaded canonical complex, built upon RNE tetramer (*Callaghan et al., 2005*), in which each RNE monomer interacts with one RhlB dimer, one enolase dimer, and one PNPase trimer. According to the amino acid sequence or reported mass in the literature (for enolase; *Spring and Wold, 1971*), the masses are as follows.

RNE monomer = 118 kDa
RhlB monomer = 47.1 kDa
Enolase monomer = 45.7 kDa
PNPase monomer = 77.1 kDa

Considering a linker (five to seven amino acids depending on the strain, or 0.56–0.78 kDa) and mEos3.2 (27.2 kDa based on the sequence), we estimate the effective mass of WT RNE-mEos3.2 complex to be ~2.27 MDa. Other strains used in this study (such as LacY variants or MTS) were calculated based on the amino acid sequence including a linker and mEos3.2.

# Appendix 3

**Appendix 3—key resources table**

| Reagent type (species) or resource | Designation | Source or reference | Identifiers | Additional information |
|---|---|---|---|---|
| Strain, strain background (*Escherichia coli*) | SK1 | *Bachmann, 1972* | | *E. coli* MG1655 wild-type |
| Genetic reagent (*Escherichia coli*) | SK47 | *Sanamrad et al., 2014* | | BW25993 *rplA::rplA-mEos2* |
| Genetic reagent (*Escherichia coli*) | SK52 | Jacobs-Wagner | CJW6557 | MG1655 Δ*araFGH araE*::P13-*araE* |
| Genetic reagent (*Escherichia coli*) | SK72 | *Strahl et al., 2015* | | NCM3416 *rne::rne-mCherry* FRT-*cat*-FRT |
| Genetic reagent (*Escherichia coli*) | SK98 | *Kim et al., 2019* | | MG1655 Δ*lacYA* |
| Genetic reagent (*Escherichia coli*) | SK105 | *Kim et al., 2019* | CJW6643 | MG1655 Δ*lacIZYA* |
| Genetic reagent (*Escherichia coli*) | SK107 | *Thappeta et al., 2024* | CJW5685 | MG1655 *rne::rne(ΔMTS)-mCherry* |
| Genetic reagent (*Escherichia coli*) | SK186 | Jacobs-Wagner | JRH474 | MG1655 *rne::rne(1-592)-yfp* FRT-*kan*-FRT |
| Genetic reagent (*Escherichia coli*) | SK187 | Jacobs-Wagner | JRH475 | MG1655 *rne::rne-mEos3.2* FRT-*kan*-FRT |
| Genetic reagent (*Escherichia coli*) | SK213 | *Xiang et al., 2021* | CJW5158 | BW25113 *hupA::hupA-mcherry* FRT-*kan*-FRT |
| Genetic reagent (*Escherichia coli*) | SK249 | This study | | MG1655 *rne::rne*Δ*MTS-mEos3.2* FRT-*kan*-FRT See **Supplementary file 2** |
| Genetic reagent (*Escherichia coli*) | SK290 | This study | | MG1655 *rne::rne-mEOS3.2* See **Supplementary file 2** |
| Genetic reagent (*Escherichia coli*) | SK292 | This study | | MG1655 *lacYA::lacY-mEos3.2* FRT-*kan*-FRT See **Supplementary file 2** |
| Genetic reagent (*Escherichia coli*) | SK304 | This study | | MG1655 *rne::rne-mEos3.2,* Δ*rhlB*::FRT-*kan*-FRT See **Supplementary file 2** |
| Genetic reagent (*Escherichia coli*) | SK308 | This study | | MG1655 *rne::rne-mEos3.2,* Δ*pnp*::FRT-*kan*-FRT See **Supplementary file 2** |
| Genetic reagent (*Escherichia coli*) | SK360 | This study | | MG1655 Δ*araFGH araE*::P13-*araE araBAD::rne-yfp-kan rne::rne-mcherry* FRT-*cat*-FRT See **Supplementary file 2** |
| Genetic reagent (*Escherichia coli*) | SK364 | This study | | MG1655 Δ*araFGH araE*::P13-*araE araBAD::rne-yfp rne::rne-mcherry* See **Supplementary file 2** |
| Genetic reagent (*Escherichia coli*) | SK370 | *Kim et al., 2024* | | MG1655 Δ(*lacYA*) *rne::rne(1-592)-yfp* FRT-*kan*-FRT |
| Genetic reagent (*Escherichia coli*) | SK373 | This study | | MG1655 *rne::rne(1-529)-mEos3.2* FRT-*kan*-FRT See **Supplementary file 2** |
| Genetic reagent (*Escherichia coli*) | SK374 | This study | | MG1655 *rne::rne(1-592)-mEos3.2* FRT-*kan*-FRT See **Supplementary file 2** |
| Genetic reagent (*Escherichia coli*) | SK384 | This study | | MG1655 *rne::rne(1-592)-yfp* See **Supplementary file 2** |

*Appendix 3 Continued on next page*

*Appendix 3 Continued*

| Reagent type (species) or resource | Designation | Source or reference | Identifiers | Additional information |
|---|---|---|---|---|
| Genetic reagent (*Escherichia coli*) | SK394 | This study | | MG1655 Δ*lacYA* Δ*araFGH araE*::P13-*araE araBAD*::*rne-yfp rne*::*rne-mcherry*<br>See **Supplementary file 2** |
| Genetic reagent (*Escherichia coli*) | SK404 | This study | | MG1655 *rne*::*rne*(1-564)-*lacY-mEos3.2* FRT-*kan*-FRT<br>See **Supplementary file 2** |
| Genetic reagent (*Escherichia coli*) | SK405 | This study | | MG1655 Δ(*lacYA*) *rne*::*rne*(1-564)-*lacY-mEos3.2* FRT-*kan*-FRT<br>See **Supplementary file 2** |
| Genetic reagent (*Escherichia coli*) | SK407 | This study | | MG1655 *lacZYA*::*lacZ-mEos3.2* FRT-*kan*-FRT<br>See **Supplementary file 2** |
| Genetic reagent (*Escherichia coli*) | SK411 | This study | | MG1655 *rne*::*rne-mEos3.2*<br>pUC19-lacI-lacZonly (amp)<br>See **Supplementary file 2** |
| Genetic reagent (*Escherichia coli*) | SK424 | This study | | MG1655 *lacYA*::*lacY*(1-73)-*mEos3.2* FRT-*kan*-FRT<br>See **Supplementary file 2** |
| Genetic reagent (*Escherichia coli*) | SK425 | This study | | MG1655 *lacYA*::*lacY*(1-192)-*mEos3.2* FRT-*kan*-FRT<br>See **Supplementary file 2** |
| Genetic reagent (*Escherichia coli*) | SK455 | This study | | MG1655 Δ*lacIZYA* pUC19-lacI-plac-mEos3.2-MTS (amp)<br>See **Supplementary file 2** |
| Genetic reagent (*Escherichia coli*) | SK466 | This study | | MG1655 *rne*::*rne*(1-564)-*lacY*(1-73)-*rne*CTD-*mEos3.2* FRT-*kan*-FRT<br>See **Supplementary file 2** |
| Genetic reagent (*Escherichia coli*) | SK467 | This study | | MG1655 *rne*::*rne*(1-564)-*lacY*(1-192)-*rne*CTD-*mEos3.2* FRT-*kan*-FRT<br>See **Supplementary file 2** |
| Genetic reagent (*Escherichia coli*) | SK482 | This study | | MG1655 *rne*::*rne-venus hupA*::*hupA-mcherry* FRT-*kan*-FRT<br>See **Supplementary file 2** |
| Genetic reagent (*Escherichia coli*) | SK486 | This study | | MG1655 *rne*::*rne*(1-592)-*venus hupA*::*hupA-mcherry* FRT-*kan*-FRT<br>See **Supplementary file 2** |
| Genetic reagent (*Escherichia coli*) | SK505 | This study | | MG1655 Δ(*lacYA*) *rne*::*rne*(1-564)-*lacY*(1-73)-*rne*CTD-*mEos3.2* FRT-*kan*-FRT<br>See **Supplementary file 2** |
| Genetic reagent (*Escherichia coli*) | SK506 | This study | | MG1655 Δ(*lacYA*) *rne*::*rne*(1-564)-*lacY*(1-192)-*rne*CTD-*mEos3.2* FRT-*kan*-FRT<br>See **Supplementary file 2** |
| Genetic reagent (*Escherichia coli*) | SK507 | This study | | MG1655 *rne*::*rne*(1-564)-*lacY*(1-73)-*mEos3.2* FRT-*kan*-FRT<br>See **Supplementary file 2** |
| Genetic reagent (*Escherichia coli*) | SK508 | This study | | MG1655 Δ(*lacYA*) *rne*::*rne*(1-564)-*lacY*(1-73)-*mEos3.2* FRT-*kan*-FRT<br>See **Supplementary file 2** |
| Genetic reagent (*Escherichia coli*) | SK512 | This study | | MG1655 *rne*::*rne-mEos3.2 hupA*::*hupA-mcherry* kan<br>See **Supplementary file 2** |
| Genetic reagent (*Escherichia coli*) | SK592 | This study | | MG1655 *rne*::*rne*(1-564)-*lacY*(1-192)-*mEos3.2* FRT-*kan*-FRT<br>See **Supplementary file 2** |
| Genetic reagent (*Escherichia coli*) | SK593 | This study | | MG1655 Δ(*lacYA*) *rne*::*rne*(1-564)-*lacY*(1-192)-*mEos3.2* FRT-*kan*-FRT<br>See **Supplementary file 2** |

*Appendix 3 Continued*

| Reagent type (species) or resource | Designation | Source or reference | Identifiers | Additional information |
|---|---|---|---|---|
| Genetic reagent (*Escherichia coli*) | SK594 | This study | | MG1655 Δ(*lacYA*) *rne::rne*(1-592)-*yfp* <br> See **Supplementary file 2** |
| Genetic reagent (*Escherichia coli*) | SK595 | This study | | MG1655 Δ(*lacYA*) *rne::rne-mEos3.2* FRT-*kan*-FRT <br> See **Supplementary file 2** |
| Genetic reagent (*Escherichia coli*) | SK598 | This study | | MG1655 Δ(*lacYA*) *rne::rne*(1-564)-*lacY-rne*CTD-*mEos3.2* FRT-*kan*-FRT <br> See **Supplementary file 2** |
| Genetic reagent (*Escherichia coli*) | SK741 | This study | | MG1655 Δ(*lacYA*) *rne::rne*(1-564)-MTS(F574AF575A)-*rne*CTD -*mEos3.2* FRT-*kan*-FRT <br> See **Supplementary file 2** |
| Genetic reagent (*Escherichia coli*) | SK742 | This study | | MG1655 Δ(*lacYA*) *rne::rne*(1-564)-MTS(F575E)-*rne*CTD-*mEos3.2* FRT-*kan*-FRT <br> See **Supplementary file 2** |
| Genetic reagent (*Escherichia coli*) | SK743 | This study | | MG1655 Δ(*lacYA*) *rne::rne*(1-564)-MTS(F582E)-*rne*CTD-*mEos3.2* FRT-*kan*-FRT <br> See **Supplementary file 2** |
| Genetic reagent (*Escherichia coli*) | SK748 | This study | | MG1655 Δ(*lacYA*) *rne::rne*(1-564)-MTS(F574AF575A)-*mEos3.2* FRT-*kan*-FRT <br> See **Supplementary file 2** |
| Genetic reagent (*Escherichia coli*) | SK749 | This study | | MG1655 Δ(*lacYA*) *rne::rne*(1-564)-MTS(F575E)-*mEos3.2* FRT-*kan*-FRT <br> See **Supplementary file 2** |
| Genetic reagent (*Escherichia coli*) | SK750 | This study | | MG1655 Δ(*lacYA*) *rne::rne*(1-564)-MTS(F582E)-*mEos3.2* FRT-*kan*-FRT <br> See **Supplementary file 2** |
| Recombinant DNA reagent | pBAD18Kan (plasmid) | **Guzman et al., 1995** | | |
| Recombinant DNA reagent | pET29b-H6_ Streptavidin _sfGFP (plasmid) | Addgene | Plasmid # 124296 | A gift from Mark Arbing (Addgene plasmid # 124296; http://n2t.net/addgene: 124296; RRID:Addgene_124296) |
| Recombinant DNA reagent | pKD13 (plasmid) | **Datsenko and Wanner, 2000** | | |
| Recombinant DNA reagent | pUC19 (plasmid) | New England Biolabs | N3041S | |
| Recombinant DNA reagent | SJK1606 (plasmid) | This study | | pBAD18kan-mEos3.2-MTS <br> See **Supplementary file 2** |
| Recombinant DNA reagent | SJK1689 (plasmid) | This study | | pUC19-lacI-lacY2-CTD-mEos3.2-Kan <br> See **Supplementary file 2** |
| Recombinant DNA reagent | SJK1697 (plasmid) | This study | | pUC19-lacI-lacY6-CTD-mEos3.2-frtKanfrt <br> See **Supplementary file 2** |
| Recombinant DNA reagent | SJK1716 (plasmid) | This study | | pUC19-lacI-lacY12-CTD-mEos3.2-frtKanfrt <br> See **Supplementary file 2** |
| Recombinant DNA reagent | SK141 (plasmid) | **Kim et al., 2019** | CJW6647 | pUC19-lacI-lacZonly |
| Recombinant DNA reagent | SK189 (plasmid) | Jacobs-Wagner | JRH515 | pBAD18 rne-yfp-kan |
| Recombinant DNA reagent | SK567 (plasmid) | This study | | pET29b-H6_Streptavidin_mEos3.2 <br> See **Supplementary file 2** |
| Other | Glass slides | Fisher | 12-544-1 | |

*Appendix 3 Continued on next page*

*Appendix 3 Continued*

| Reagent type (species) or resource | Designation | Source or reference | Identifiers | Additional information |
|---|---|---|---|---|
| Other | #1.5 coverslips | Fisher Or VWR | 12544A 16004-344 | |
| Chemical compound, drug | Agarose | Invitrogen | 16500-100 | |
| Chemical compound, drug | Glycerol | Invitrogen | 15514011 | |
| Chemical compound, drug | Casamino acids | Bacto | 223050 | |
| Chemical compound, drug | Thiamine | Research Products International | T21020 | |
| Chemical compound, drug | Rifampicin | Sigma-Aldrich | R3501 | 250MG |
| Chemical compound, drug | Chloramphenicol | Acros organics | 227920250 | |
| Chemical compound, drug | Isopropyl-β-thiogalactopyranoside (IPTG) | Thermo Fisher Scientific | C9H18O5S | |
| Chemical compound, drug | L(+)-Arabinose | Acros organics | 365180250 | |
| Software algorithm | Oufti | *Paintdakhi et al., 2016* | | https://oufti.org/ |
| Software algorithm | u-track | *Jaqaman et al., 2008* | | https://github.com/DanuserLab/u-track |
| Software algorithm | spotNorm | This study | | https://github.com/sjkimlab/2025_RNaseE |
| Software algorithm | NAMD 3.0 | *Phillips et al., 2005* | | https://www.ks.uiuc.edu/Research/namd/ |
| Software algorithm | CHARMM-GUI | *Jo et al., 2008* | | https://www.charmm-gui.org/ |
| Sequence-based reagent | lacZ530F | *Kim et al., 2024* | PCR primers | TTTTACGCGCCGGAGAAAAC |
| Sequence-based reagent | lacZ530R | *Kim et al., 2024* | PCR primers | AGTCGGTTTATGCAGCAACG |
| Sequence-based reagent | lacZ2732F | *Kim et al., 2024* | PCR primers | TTACTGCCGCCTGTTTTGAC |
| Sequence-based reagent | lacZ2732R | *Kim et al., 2024* | PCR primers | TGTAGCGGCTGATGTTGAAC |
| Sequence-based reagent | gapA274F | *Kim et al., 2024* | PCR primers | GTTGTCGCTGAAGCAACTGG |
| Sequence-based reagent | gapA274R | *Kim et al., 2024* | PCR primers | CGATGTCCTGGCCAGCATAT |

