## [Editor Report · eLife Assessment]

This **valuable** study uses single-molecule imaging to characterize factors controlling the localization, mobility, and function of RNase E in *E. coli*, a key bacterial ribonuclease central to mRNA catabolism. The supporting evidence for the differential roles of RNAse E's membrane targeting sequence (MTS) and the C-terminal domain (CTD) to RNAse E's diffusion and membrane association is **convincing**. It provides insight into how RNAse E shapes the spatiotemporal organization of RNA processing in bacterial cells. This interdisciplinary work will be of interest to cell biologists, microbiologists, biochemists, and biophysicists.

---

## [Referee Report · Reviewer #1 (Public review)]

This paper by Troyer et al. measures the positioning and diffusivity of RNaseE-mEos3.2 proteins in *E. coli* as a function of rifampicin treatment, compares RNaseE to other *E. coli* proteins, and measures the effect of changes in domain composition on this localization and motion. The straightforward study is thoroughly presented, including very good descriptions of the imaging parameters and the image analysis/modeling involved, which is good because the key impact of the work lies in presenting this clear methodology for determining the position and mobility of a series of proteins in living bacteria cells.

Most of my concerns in the original review were addressed in this round of revisions based on new text, experiments, and analysis, including most notably:

-A revision of the abstract to focus on the actual topic of the manuscript.

-New experiments (Fig. S1) to confirm that there is no significant undercounting of the fast-moving cytoplasmic population

-Removing the experiments discussion related to degradosome proteins rather than overstating results.

-Improving the logical flow and writing.

One minor concern still remains:

-Though the discussion of the rifampicin-treated cells is improved, this experiment is motivated (line 196) as "To test the effect of mRNA substrates on RNE diffusion", but the conclusion of the paragraph (based on similarities with the effect on LacY) is that the observed changes are due to factors other than the concentration of mRNA substrates, such that the effect of mRNA has not been tested.

---

## [Referee Report · Reviewer #2 (Public review)]

Summary:

Troyer and colleagues have studied the in vivo localisation and mobility of the *E. coli* RNaseE (a protein key for mRNA degradation in all bacteria) as well as the impact of two key protein segments (MTS and CTD) on RNase E cellular localisation and mobility. Such sequences are important to study since there is significant sequence diversity within bacteria, as well as lack of clarity about their functional effects. Using single-molecule tracking in living bacteria, the authors confirmed that >90% of RNaseE localised on the membrane, and measured its diffusion coefficient. Via a series of mutants, they also showed that MTS leads to stronger membrane association and slower diffusion compared to a transmembrane motif (despite the latter being more embedded in the membrane), and that the CTD weakens membrane binding. The study also rationalised how the interplay of MTS and CTD modulate mRNA metabolism (and hence gene expression) in different cellular contexts.

The authors have also done an excellent job addressing reviewer's concerns and improving the manuscript during revision.

---

## [Referee Report · Reviewer #3 (Public review)]

Summary:

The manuscript by Troyer et al quantitatively measured the membrane localization and diffusion of RNase E, an essential ribonuclease for mRNA turnover as well as tRNA and rRNA processing in bacteria cells. Using single-molecule tracking in live *E. coli* cells, the authors investigated the impact of membrane targeting sequence (MTS) and the C-terminal domain (CTD) on the membrane localization and diffusion of RNase E under various perturbations. Finally, the authors tried to correlate the membrane localization of RNase E to its function on co- and post-transcriptional mRNA decay using lacZ mRNA as a model.

The major findings of the manuscripts include:

(1) WT RNase E is mostly membrane localized via MTS, confirming previous results. The diffusion of RNase E is increased upon removal of MTS or CTD, and more significantly increased upon removal of both regions.

(2) By tagging RNase E MTS and different lengths of LacY transmembrane domain (LacY2, LacY6 or LacY12) to mEos3.2, the results demonstrate that short LacY transmembrane sequence (LacY2 and LacY6) can increase the diffusion of mEos3.2 on the membrane compared to MTS, further supported by the molecular dynamics simulation. The similar trend was roughly observed in RNase E mutants with MTS switched to LacY transmembrane domains.

(3) The removal of RNase E MTS significantly increases the co-transcriptional degradation of lacZ mRNA, but has minimal effect on the post-transcriptional degradation of lacZ mRNA. Removal of CTD of RNase E overall decrease the mRNA decay rates, suggesting the synergistic effect of CTD on RNase E activity.

Strengths:

(1) The manuscript is clearly written with very detailed methods description and analysis parameters.

(2) The conclusions are mostly supported by the data and analysis.

(3) Some of the main conclusions are interesting and important for understanding the cellular behavior and function of RNase E.

Weaknesses:

The authors have addressed my previous concerns in the revised manuscript.

Comments on revisions:

I have one additional comment. When interpreting the small increase in the diffusion coefficient of RNase E when treating the cell with rifampicin, the authors rule out the possibility that only a small fraction of RNase E interacts with mRNA and suggest that it is more likely the mRNA-RNase E interaction is transient. However, I am wondering about an alternative possibility that RNase E prefers mRNAs with low ribosome density or even untranslated mRNAs?

---

## [Author Response]

The following is the authors’ response to the original reviews.

**Reviewer #1 (Public review):**
This paper measures the positioning and diffusivity of RNaseE-mEos3.2 proteins in *E. coli* as a function of rifampicin treatment, compares RNaseE to other *E. coli* proteins, and measures the effect of changes in domain composition on this localization and motion. The straightforward study is thoroughly presented, including very good descriptions of the imaging parameters and the image analysis/modeling involved, which is good because the key impact of the work lies in presenting this clear methodology for determining the position and mobility of a series of proteins in living bacteria cells.

Thank you for the nice summary and positive feedback on the descriptions and methodology.

My key notes and concerns are listed below; the most important concerns are indicated with asterisks.(1) The very start of the abstract mentions that the domain composition of RNase E varies among species, which leads the reader to believe that the modifications made to *E. coli* RNase E would be to swap in the domains from other species, but the experiment is actually to swap in domains from other *E. coli* proteins. The impact of this work would be increased by examining, for instance, RNase E domains from *B. subtilis* and C. crescentus as mentioned in the introduction.

Thank you for the suggestions. We agree that the sentence may convey an unintended expectation. Our original intention was to note the presence and absence of certain domains of RNase E (e.g. membrane-binding motif and CTD) vary across species, rather than the actual sequence variations. To avoid any misinterpretation, we decided to remove the sentence from the abstract. Using the domains of *B. subtilis* and C. crescentus RNase E in *E. coli* is a very interesting suggestion, but we will leave that for a future study.

(2) Furthermore, the introduction ends by suggesting that this work will modulate the localization, diffusion, and activity of RNase E for "various applications", but no applications are discussed in the discussion or conclusion. The impact of this work would be increased by actually indicating potential reasons why one would want to modulate the activity of RNase E.

Thank you for this suggestion. For example, an *E. coli* strain expressing membranebound RNase E without CTD can help stabilize mRNAs and enhance protein expression. In fact, this idea was used in a commercial BL21 cell line (Invitrogen’s One Shot BL21 Star), to increase the yield of protein expression. We also think that environmentally modulated MB% of RNase E can be useful for controlling the mRNA half-lives and protein expression levels in different conditions. We discussed these ideas at the end of the Discussion.

(3) Lines 114 - 115: "The xNorm histogram of RNase E shows two peaks corresponding to each side edge of the membrane": "side edge" is not a helpful term. I suggest instead: "...corresponding to the membrane at each side of the cell"

Thank you. We made the suggested change.

(4) A key concern of this reviewer is that, since membrane-bound proteins diffuse more slowly than cytoplasmic proteins, some significant undercounting of the % of cytoplasmic proteins is expected due to decreased detectability of the faster-moving proteins. This would not be a problem for the LacZ imaging where essentially all proteins are cytoplasmic, but would significantly affect the reported MB% for the intermediate protein constructs. How is this undercounting considered and taken into account? One could, for instance, compare LacZ vs. LacY (or RNase E) copy numbers detected in fixed cells to those detected in living cells to estimate it.

Thank you for raising this point and suggesting a possible way to address this. We compared the number of tracks for mEos3.2-fused proteins in live vs fixed cells and tested the undercounting effect of cytoplasmic molecules. We compared WT RNase E molecules in live and fixed cells and found that there are about 50% lower molecules detected in the fixed cells, which agrees with the expectation that fluorescent proteins lose their signal upon fixation. Similarly, cytoplasmic RNase E (RNase E ΔMTS) copy number was also ~50% less in the fixed cells compared to live cells. If cytoplasmic molecules were undercounted compared membrane-bound molecules in live cells, fixation would reduce the copy number less than 50%. The comparable ratio of 50% indicates that the undercounting issue is not significant. This control analysis is provided in Figure S1B-C, and we made corresponding textual change in the result section as below:

For this analysis, we first confirmed that proteins localized on the membrane and in the cytoplasm are detected with equal probability, despite differences in their mobilities (Fig. S1B-C).

(5) The rifampicin treatment study is not presented well. Firstly, it is found that LacY diffuses more rapidly upon rifampicin treatment. This change is attributed to changes in crowding at the membrane due to mRNA. Several other things change in cells after adding rif, including ATP levels, and these factors should be considered. More importantly, since the change in the diffusivity of RNaseE is similar to the change in diffusivity of LacY, then it seems that most of the change in RNaseE diffusion is NOT due to RNaseE-mRNAribosome binding, but rather due to whatever crowding/viscosity effects are experienced by LacY (along these lines: the error reported for D is SEM, but really should be a confidence interval, as in Figure 1, to give the reader a better sense of how different (or similar) 1.47 and 1.25 are).

We agree with the reviewer that upon rifampicin treatment, RNase E’s D increases to a similar extent as that of LacY. Hence, the increase likely arises from a factor common to both proteins. We have added the reviewer’s suggested interpretation as a possible explanation in the manuscript as below.

The similar fold change in D_RNE_ and D_LacY_ upon rif treatment suggests that the change in RNE diffusion may largely be attributed to physical changes in the intracellular environment (such as reduced viscosity or macromolecular crowding[41,42]), rather than a loss of RNA-RNE interactions.

As requested by the reviewer, we have provided confidence intervals for our D values in Table S8. Because these intervals are very narrow, we chose to present the SEM as the error metric for D and have also reported the corresponding errors for the fold-change values whenever we describe the fold differences between D values.

(6) Lines 185-189: it is surprising to me that the CTD mutants both have the same change in D (5.5x and 5.3x) relative to their full-length counterparts since D for the membranebound WT protein should be much less sensitive to protein size than D for the cytoplasmic MTS mutant. Can the authors comment?

Perhaps the reviewer understood that these differences are the ratios between +/-CTD (e.g. WT RNE vs ΔCTD). However, the differences we mentioned were from membrane-bound vs cytoplasmic versions of RNase E with comparable sizes (e.g. WT RNase E vs RNase E ΔMTS). We modified text and added a summary sentence at the end of the paragraph to clarify the point.

We found that D_ΔMTS_ is ~5.5 times that of D_RNE_ (Fig. 3B). [...] Together, these results suggest that the membrane binding reduces RNE mobility by a factor of 5.

That being said, we also realized a similar fold difference between +/-CTD. Specifically, WT RNE vs RNE ΔCTD (both membrane-bound) show a ~4.1-fold difference and RNE ΔMTS vs RNE ΔMTS ΔCTD (both cytoplasmic) show ~3.9-fold difference. We do not currently do not have a clear explanation for this pattern. Given that these two pairs have a similar change in mass, we speculate that the relationship between D and molecular mass may be comparable for membrane-bound and free-floating RNE variants.

(7) Lines 190-194. Again, the confidence intervals and experimental uncertainties should be considered before drawing biological conclusions. It would seem that there is "no significant change" in the rhlB and pnp mutants, and I would avoid saying "especially for ∆pnp" when the same conclusion is true for both (one shouldn't say 1.04 is "very minute" and 1.08 is just kind of small - they are pretty much the same within experiments like this).

Thank you for raising this point, which we fully agree with. That being said, we decided to remove results related to the degradosome proteins to improve the flow of the paper. We are preparing another paper related to the RNA degradosome complex formation.

(8) Lines 221-223 " This is remarkable because their molecular masses (and thus size) are expected to be larger than that of MTS" should be reconsidered: diffusion in a membrane does not follow the Einstein law (indeed lines 223-225 agree with me and disagree with lines 221-223). (Also the discussion paragraph starting at line 375). Rather, it is generally limited by the interactions with the transmembrane segments with the membrane. So Figure 3D does not contain the right data for a comparison, and what is surprising to me is that MTS doesn't diffuse considerably faster than LacY2.

We agree with the reviewer’s point that diffusion in a membrane does not follow the Stokes-Einstein law. That is why we introduced Saffman’s model. However, even in this model, proteins of larger size (or mass) should be slower than smaller size (a reason why we presented Figure 3D, now 4D). In other words, both Einstein and Saffman models predict that larger particles diffuse slower, although the exact scaling relationship differs between two models. Here, we assume that mass is related to the size. Contrary to Saffman’s model for membrane proteins, LacY2 diffuses faster than MTS despite of large size. Using MD simulations, we showed that this discrepancy can be explained by different interaction energies as the reviewer mentioned. This analysis further demonstrates that the size is not the only factor to consider protein diffusion in the membrane. We edited the paragraph to clarify the expectations and our interpretations.

According to the Stokes-Einstein relation for diffusion in simple fluids[49] and the Saffman-Delbruck diffusion model for membrane proteins, D decreases as particle size increases, albeit with different scaling behaviors. […] Thus, if size (or mass) were the primary determinant of diffusion, LacY2 and LacY6 would diffuse more slowly than the smaller MTS. The observed discrepancy instead implies that D may be governed by how each motif interacts with the membrane. For example, the way that TM domains are anchored to the membrane may facilitate faster lateral diffusion with surrounding lipids.

(9) The logical connection between the membrane-association discussion (which seems to ignore associations with other proteins in the cell) and the preceding +/- rifampicin discussion (which seeks to attribute very small changes to mRNA association) is confusing.

Thank you for raising this point. We re-arranged the second result section to present diffusion due to membrane binding first before rifampicin. Furthermore, we stated our hypothesis and expectations in the beginning of the results section. This addition will legitimate our logic flow.

(10) Separately, the manuscript should be read through again for grammar and usage. For instance, the title should be: "Single-molecule imaging reveals the *roles* of *the* membrane-binding motif and *the* C-terminal domain of RNase E in its localization and diffusion in *Escherichia coli*". Also, some writing is unwieldy, for instance, "RNase E's D" would be easier to read if written as D_{RNaseE}. (underscore = subscript), and there is a lot of repetition in the sentence structures.

Thank you for catching grammar mistakes. We went through extensive proofreading to avoid these mistakes and also used simple notation suggested by the reviewer, such as D_RNE_, to make it easier to read. Thank you again for your suggestions.

**Reviewer #2 (Public review):**
Summary:Troyer and colleagues have studied the in vivo localisation and mobility of the *E. coli* RNaseE (a protein key for mRNA degradation in all bacteria) as well as the impact of two key protein segments (MTS and CTD) on RNase E cellular localisation and mobility. Such sequences are important to study since there is significant sequence diversity within bacteria, as well as a lack of clarity about their functional effects. Using single-molecule tracking in living bacteria, the authors confirmed that >90% of RNaseE localised on the membrane, and measured its diffusion coefficient. Via a series of mutants, they also showed that MTS leads to stronger membrane association and slower diffusion compared to a transmembrane motif (despite the latter being more embedded in the membrane), and that the CTD weakens membrane binding. The study also rationalised how the interplay of MTS and CTD modulate mRNA metabolism (and hence gene expression) in different cellular contexts.Strengths:The study uses powerful single-molecule tracking in living cells along with solid quantitative analysis, and provides direct measurements for the mobility and localisation of *E. coli* RNaseE, adding to information from complementary studies and other bacteria. The exploration of different membrane-binding motifs (both MTS and CTD) has novelty and provides insight on how sequence and membrane interactions can control function of protein-associated membranes and complexes. The methods and membrane-protein standards used contribute to the toolbox for molecular analysis in live bacteria.

Thank you for the nice summary of our work and positive comments about the paper’s strengths.

Weaknesses:The Results sections can be structured better to present the main hypotheses to be tested. For example, since it is well known that RNase E is membrane-localised (via its MTS), one expects its mobility to be mainly controlled by the interaction with the membrane (rather than with other molecules, such as polysomes and the degradosome). The results indeed support this expectation - however, the manuscript in its current form does not lay down the dominant hypothesis early on (see second Results chapter), and instead considers the rifampicin-addition results as "surprising"; it will be best to outline the most likely hypotheses, and then discuss the results in that light.

Thank you for this comment. We addressed this point by stating our main hypothesis from the beginning of the results section. We also agree with the reviewer that the membrane binding effect should be discussed first; hence, we re-arranged the result section. In the revised manuscript, we discuss the effect of membrane binding on diffusion first, followed by rif effects.

Similarly, the authors should first discuss the different modes of interaction for a peripheral anchor vs a transmembrane anchor, outline the state of knowledge and possibilities, and then discuss their result; in its current version, the ms considers the LacY2 and LacY6 faster diffusion compared to MTS "remarkable", but considering the very different mode of interaction, there is no clear expectation prior to the experiment. In the same section, it would be good to see how the MD simulations capture the motion of LacY6 and LacY12, since this will provide a set of results consistent with the experimental set.

Thank you for pointing this out. In fact, there is little discussion in the literature about the different modes of interaction for a peripheral anchor vs a transmembrane anchor. To our knowledge, our work (experiments and MD simulations) is the first that directly compared the two to reveal that the peripheral anchor has higher interaction energy than the transmembrane anchor. We added a sentence “Despite the prevalence of peripheral membrane proteins, how they interact with the membrane and how this differs from TM proteins remain poorly understood”. Furthermore, we added the MD simulation result of LacY6 and LacY12 in Figure 4E-F.

The work will benefit from further exploration of the membrane-RNase E interactions; e.g., the effect of membrane composition is explored by just using two different growth media (which on its own is not a well-controlled setting), and no attempts to change the MTS itself were made. The manuscript will benefit from considering experiments that explore the diversity of RNaseE interactions in different species; for example, the authors may want to consider the possibility of using the membrane-localisation signals of functional homologs of RNaseE in different bacteria (e.g., *B. subtilis*). It would be good to look at the effect of CTD deletions in a similar context (i.e., in addition to the MTS substitution by LacY2 and LacY6).

Thank you very much for this suggestion. During revision, we engineered point mutations in MTS and analyzed critical hydrophobic residues for membrane binding. We characterized MB% in both +/-CTD variants (Fig. 2 and Fig. S6) and their effect on lacZ mRNA degradation (Fig. 6). We will leave the use of membrane motif of *B. subtilis* RNase E for future study.

The manuscript will benefit from further discussion of the unstructured nature of the CTD, especially since the RNase CTD is well known to form condensates in Caulobacter crescentus; it is unclear how the authors excluded any roles for RNaseE phase separation in the mobility of RNaseE in *E. coli* cells.

Yes, we agree with the reviewer that the intrinsically disordered nature of the CTD might contribute to condensate formation. We explored this possibility using both epifluorescence microscopy (with a YFP fusion) and single-molecule imaging with cluster analysis (using an mEos3.2 fusion). Please see Figure S8. We did observe some weak de-clustering of RNase E upon CTD deletion. In the current study, we are unable to quantify the extent to which clustering contributes to the slow diffusion of RNase E. However, we speculate that the clustering may be linked to the low MB% of certain RNE mutants containing CTD, and we discussed this possibility in the Discussion.

[…] further supporting that the CTD decreases membrane association across RNE variants. We speculate that this effect may be related to the CTD’s role in promoting phase-separated ribonucleoprotein condensates, as observed in Caulobacter crescentus[19]. In *E. coli*, we also observed a modest increase in the clustering tendency of RNE compared to ΔCTD (Fig. S8).

Some statements in the Discussion require support with example calculations or toning down substantially. Specifically, it is not clear how the authors conclude that RNaseE interacts with its substrate for a short time (and what this time may actually be); further, the speculation about the MTS "not being an efficient membrane-binding motif for diffusion" lacks adequate support as it stands.

Thank you for these points. To elaborate our point on transient interaction between RNase E and RNA, we added a sentence “Specifically, if RNE interacts with mRNAs for ~20 ms or less, the slow-diffusing state would last shorter than the frame interval and remain undetected in our experiment.” Also, we added this sentence in the discussion.

One possible explanation is that RNA-bound RNE (and RNase Y) is short-lived compared to our frame interval (~20 ms), unlike other RNA-binding proteins related to transcription and translation, interacting with RNA for ~1 min for elongation [48].

Plus, we clarified the wording used in the second sentence that the reviewer pointed out as follows,

Lastly, the slow diffusion of the MTS in comparison to LacY2 and LacY6 suggests that MTS is less favorable for rapid lateral motion in the membrane.

**Reviewer #3 (Public review):**
Summary:The manuscript by Troyer et al quantitatively measured the membrane localization and diffusion of RNase E, an essential ribonuclease for mRNA turnover as well as tRNA and rRNA processing in bacteria cells. Using single-molecule tracking in live *E. coli* cells, the authors investigated the impact of membrane targeting sequence (MTS) and the Cterminal domain (CTD) on the membrane localization and diffusion of RNase E under various perturbations. Finally, the authors tried to correlate the membrane localization of RNase E to its function on co- and post-transcriptional mRNA decay using lacZ mRNA as a model.The major findings of the manuscripts include:(1) WT RNase E is mostly membrane localized via MTS, confirming previous results. The diffusion of RNase E is increased upon removal of MTS or CTD, and more significantly increased upon removal of both regions.(2) By tagging RNase E MTS and different lengths of LacY transmembrane domain (LacY2, LacY6, or LacY12) to mEos3.2, the results demonstrate that short LacY transmembrane sequence (LacY2 and LacY6) can increase the diffusion of mEos3.2 on the membrane compared to MTS, further supported by the molecular dynamics simulation. A similar trend was roughly observed in RNase E mutants with MTS switched to LacY transmembrane domains.(3) The removal of RNase E MTS significantly increases the co-transcriptional degradation of lacZ mRNA, but has minimal effect on the post-transcriptional degradation of lacZ mRNA. Removal of CTD of RNase E overall decreases the mRNA decay rates, suggesting the synergistic effect of CTD on RNase E activity.Strengths:(1) The manuscript is clearly written with very detailed method descriptions and analysis parameters.(2) The conclusions are mostly supported by the data and analysis.(3) Some of the main conclusions are interesting and important for understanding the cellular behavior and function of RNase E.

Thank you for your thorough summary of our work and positive comments.

Weaknesses:(1) Some of the observations show inconsistent or context-dependent trends that make it hard to generalize certain conclusions. Those points are worth discussion at least. Examples include:(a) The authors conclude that MTS segment exhibits reduced MB% when succinate is used as a carbon source compared to glycerol, whereas LacY2 segment maintains 100% membrane localization, suggesting that MTS can lose membrane affinity in the former growth condition (Ln 341-342). However, the opposite case was observed for the WT RNase E and RNase E-LacY2-CTD, in which RNase E-LacY2-CTD showed reduced MB% in the succinate-containing M9 media compared to the WT RNase E (Ln 264-267). This opposite trend was not discussed. In the absence of CTD, would the media-dependent membrane localization be similar to the membrane localization sequence or to the fulllength RNase E?

This is a great point. Thank you for pointing out the discrepancy in data. We think the weak membrane interaction of RNaseE-lacY2-CTD likely stems from the structure instability in the presence of the CTD. Our data shows that an RNase E variant with a cytoplasmic population under a normal growth condition exhibits a greater cytoplasmic fraction in a poor growth media. In contrast, RNaseE-MTS and RNaseE-LacY2 lacking the CTD both showed 100% MB% under both normal and poor growth conditions. These results are presented in Figure S6 and further discussed in the Discussion section.

The loss of MB% in LacY2-based RNE was observed only in the presence of the CTD (Fig. S6D), suggesting that the CTD negatively affects membrane binding of RNE, possibly by altering protein conformation. In fact, all ΔCTD RNE mutants we tested exhibited higher MB% than their CTD-containing counterparts (Fig. S6A-B).

(b) When using mEos3.2 reporter only, LacY2 and LacY6 both increase the diffusion of mEos3.2 compared to MTS. However, when inserting the LacY transmembrane sequence into RNase E or RNase E without CTD, only the LacY2 increases the diffusion of RNase E. This should also be discussed.

Thank you for raising this point. As the reviewer pointed out, as the membrane motifs, both LacY2 and LacY6 diffuse faster than the MTS, but when they are fused to RNE, only LacY2-based RNE diffuses faster than MTS-based RNE. We speculate that it is possibly due to a structural reason—having four (large) LacY6 in a tetrameric arrangement may cancel out the original fast-diffusing property of LacY6. We added this idea in the result section:

This result may be due to the high TM load (24 helices) created by four LacY6 anchors in the RNE tetramer. Although all constructs are tetrameric, the 24-helix load (LacY6), compared with 8 (LacY2) and 4 (MTS), likely enlarges the membrane-embedded footprint and increases drag, thereby changing the mobility advantages assessed as standalone membrane anchors.

(2) The authors interpret that in some cases the increase in the diffusion coefficient is related to the increase in the cytoplasm localization portion, such as for the LacY2 inserted RNase E with CTD, which is rational. However, the authors can directly measure the diffusion coefficient of the membrane and cytoplasm portion of RNase E by classifying the trajectories based on their localizations first, rather than just the ensemble calculation.

Thank you for this suggestion. Currently, because of the 2D projection effect from imaging, we cannot clearly distinguish which individual tracks are from the cytoplasm or from the inner membrane based on the localization. Therefore, we are unable to assign individual tracks as membrane-bound or cytoplasmic. However, we can demonstrate that the xNorm data can be separated into two different spatial populations based on the diffusion coefficient. D. That is we can plot xNorm of slow tracks vs xNorm of fast tracks. This analysis showed that the slow tracks have LacY-like xNorm profiles while the fast tracks have LacZ-like xNorm profiles, also quantitatively supporting our MB% fitting results. We have added this analysis to Figure S2.

(3) The error bars of the diffusion coefficient and MB% are all SEM from bootstrapping, which are very small. I am wondering how much of the difference is simply due to a batch effect. Were the data mixed from multiple biological replicates? The number of biological replicates should also be reported.

Thank you for raising this point. In the original manuscript, we reported the number of tracks analyzed and noted that all data was from at least three separate biological replicates (measurements were repeated at least three different days). Furthermore, in the revised manuscript, we have provided the number of cells imaged in Table S6.

(4) Some figures lack p-values, such as Figures 4 and 5C-D. Also, adding p-values directly to the bar graphs will make it easier to read.

Thank you for checking these details. We added p values in the graphs showing k_d1_ and k_d2_ (Table S7).

**Reviewer #2 (Recommendations for the authors):**
Minor and technical points:(1) Clarity and flow will be improved if each section first highlights the objective for the experiments that are described (e.g., line 240).

Thank you for the suggestion. We addressed this point by editing the beginning of each subsection in the Results.

(2) Line 272 (and elsewhere)."1.33-times faster is wrong". The authors mean 33% faster (from 0.075 to 1, see Figure 4G), and not 133% faster. Needs fixing.

Thanks for pointing this out. We changed this as well as other incidences where we talk about the fold difference. For example, this particular incidence was changed to:

Indeed, in the absence of the CTD, we found that the D of LacY2-based RNE was 1.33 ± 0.01 times as fast as the MTS-based RNE.

(3) The authors need to consider the fitting of two species on their D population. e.g., how will a 93% - 7% split between diffusive species would have looked for the distribution in S4B? Note also the L1 profile in Fig S4C - while it is not hugely different from Figure S4B, the analysis gives a 41% amplitude for the fast-diffusing species. The 2-species analysis can also be used on some of the samples with much higher cytoplasmic components. Further, tracks that are in the more central region can be analysed to see whether the fast-diffusing species increase in amplitude.

Thank you for this comment. The D histograms of L1 and RNase E show a dominant peak at around 0.015, but L1 has a residual population in the shoulder (note the difference between L1’s experimental data and D1 fit, a yellow line in now Figure S3B). This residual shoulder population is absent in the D histogram of RNase E. We also performed two-species analysis as suggested by the reviewer and provided the result in Figure S3C. The analysis shows that the two-population fit (black line) is very close to one one-population fit (yellow line). While we agree with the reviewer that subpopulation analysis is helpful for other proteins that show <90% MB% (>10% significant cytoplasmic population). we found it useful to divide xNorm histogram into two populations based on the diffusivity (rather than doing two-population fit to the D histogram, which does not have spatial information). This analysis, shown in Figure S2, supports our MB% fit results.

(4) The authors suggest that the sequestration of RNaseE to the membrane limits its interaction with cytoplasmic mRNAs, and may increase mRNA lifetime. While this is true and supported by the authors' preprint (Ref15), it will also be good to consider (and discuss) that highly-transcribed regions are in the nucleoid periphery (and thus close to the membrane) and that ribosomes/polysomes are likewise predominantly peripheral (coregulation of transcription/translation) and membrane proximal.

This is an interesting point, which we appreciate very much. The lacZ gene, when induced, is shown to move to the nucleoid periphery (Yang et al. 2019, Nat Comm). Also, in our preprint (Ref 15), we engineered to have lacZ closer to the membrane, by translationally fusing it to lacY. However, the degradation rate of lacZ mRNA was not enhanced by the proximity to the membrane (for both k_d1_ and k_d2_). For lacZ mRNA, we mainly see the change in k_d1_ when RNE localization changes. We think it is due to the slow diffusion of the nascent mRNA (attached to the chromosome) and the slow diffusion of membrane-bound RNE, such that regardless of the location of the nascent mRNA, the degradation by the membrane-bound RNE is inefficient. Only when RNE is free diffusing in the cytoplasm, it seems to increase k_d1_ (the decay of nascent mRNAs).

**Reviewer #3 (Recommendations for the authors):**
(1) It will increase the clarity of the manuscript if the authors can provide better nomenclatures for different constructs, such as for different membrane targeting sequences fused to mEos3.2, full-length RNase E, or CDT truncated RNaseE.

Thank you for this suggestion. We agree that many constructions were discussed, and their naming can be confusing. To help with clarity, we have abbreviated RNase E as RNE throughout the text where appropriate.

(2) Line 342, Figure S7D should be cited instead of S6D.

Thank you for finding this error. We made a proper change in the revised manuscript.